# Multiplexed microfluidic screening of bacterial chemotaxis

**Michael R Stehnach[1†], Richard J Henshaw[1‡], Sheri A Floge[2], Jeffrey S Guasto[1]\***

[1]Department of Mechanical Engineering, Tufts University, Medford, United States; [2]Department of Biology, Wake Forest University, Winston-Salem, United States

**Abstract** Microorganism sensing of and responding to ambient chemical gradients regulates a myriad of microbial processes that are fundamental to ecosystem function and human health and disease. The development of efficient, high-throughput screening tools for microbial chemotaxis is essential to disentangling the roles of diverse chemical compounds and concentrations that control cell nutrient uptake, chemorepulsion from toxins, and microbial pathogenesis. Here, we present a novel microfluidic multiplexed chemotaxis device (MCD) which uses serial dilution to simultaneously perform six parallel bacterial chemotaxis assays that span five orders of magnitude in chemostimulant concentration on a single chip. We first validated the dilution and gradient generation performance of the MCD, and then compared the measured chemotactic response of an established bacterial chemotaxis system (*Vibrio alginolyticus*) to a standard microfluidic assay. Next, the MCD's versatility was assessed by quantifying the chemotactic responses of different bacteria (*Psuedoalteromonas haloplanktis, Escherichia coli*) to different chemoattractants and chemorepellents. The MCD vastly accelerates the chemotactic screening process, which is critical to deciphering the complex sea of chemical stimuli underlying microbial responses.

**\*For correspondence:**
Jeffrey.Guasto@tufts.edu

**Present address:** [†]Department of Physics, Brandeis University, Waltham, United States; [‡]Department of Civil, Environmental and Geomatic Engineering, Institute of Environmental Engineering, ETH Zürich, Zürich, Switzerland

**Competing interest:** The authors declare that no competing interests exist.

## Editor's evaluation

This manuscript presents a valuable new microfluidic tool that will allow researchers from different fields to rapidly quantify the chemotactic response of microbes to chemical gradients that have different strengths. Using planktonic bacteria, this paper convincingly shows that a multiplexed microfluidic device produces similar results to previously described microfluidic devices that generate only one gradient at a time. By performing on-chip dilutions, this device allows data for six different gradient strengths to be generated simultaneously, potentially reducing both experimental effort and biological variability.

## Introduction

Motile cells of all types navigate complex environments through the detection of and response to chemical signals via chemotaxis (*Adler, 1966*; *Berg and Brown, 1972*; *Stocker, 2012*; *Wadhams and Armitage, 2004*). This fundamental survival mechanism regulates countless biological processes, such as microbial foraging in marine environments (*Seymour et al., 2010*; *Stocker et al., 2008*) and reproduction (*Kaupp et al., 2008*). Consequently, considerable effort has been invested into the study of microbial chemotaxis (*Keegstra et al., 2022*; *Raina et al., 2019*) to better understand their chemotactic motility (*Lazova et al., 2011*), detection sensitivity (*Mao et al., 2003*), and transport (*Ford and Harvey, 2007*). Microfluidic devices have become an indispensable platform for disentangling the intricacies of microbial chemotaxis by virtue of their precise control over the chemical environment at scales relevant to swimming cells (*Keegstra et al., 2022*; *Ahmed et al., 2010*). Specifically, microfluidics have been employed to physically model a range of chemical landscapes, such as nutrient

**eLife digest** Many microorganisms such as bacteria swim to explore their fluid habitats, which range from the human digestive system to the oceans. They can detect minute traces of food, toxins and other chemicals in their environment, and – through a process called chemotaxis – respond by swimming towards or away from them. Chemical concentrations naturally decrease with distance away from their source, forming gradients. By sensing these chemical gradients, and adjusting their swimming direction accordingly, cells can locate nutrients and other resources in harsh environments as well as avoid toxins and potential predators.

Over the past 20 years, laboratory devices that manipulate minute volumes of fluid – known as microfluidics devices – have been indispensable for studying chemotaxis. They enable researchers to generate gradients of chemicals in carefully designed networks of microscopic channels, controlling the conditions that swimming cells are exposed to and mimicking their natural habitats. However, large-scale studies of chemotaxis have been limited by the sheer range of chemicals that are present at different levels in natural environments. Conventional microfluidic devices often compromise between distinguishing how individual cells behave, precise control over the chemical gradient, or the ability to execute multiple assays at the same time.

Here, Stehnach et al. designed a microfluidic device called the Multiplexed Chemotaxis Device. The device generates five streams of precise dilutions of a chemical and then uses these streams – alongside a control stream lacking the chemical – to measure chemotaxis in six different conditions at the same time. The device was tested using a well-studied bacterium, *Vibrio alginolyticus*, which is commonly found in marine environments. The device reliably examined the chemotaxis response of the population to various chemicals, was able to carry out multiple assays more rapidly than conventional devices, and can be easily applied to study the response of individual bacteria under the same conditions.

The Multiplexed Chemotaxis Device is relatively easy to manufacture using standard methods and therefore has the potential to be used for large-scale chemotaxis studies. In the future, it may be useful for screening new drugs to treat bacterial infections and to help identify food sources for communities of microbes living in marine environments.

patches (*Stocker et al., 2008*), and provide highly tunable concentration profiles (*Li Jeon et al., 2002*; *Sugiura et al., 2010*). Microfluidics have been broadly applied across microbial systems for both drug-dose response quantification (*Sugiura et al., 2010*) and infectious disease diagnostics (*Welch et al., 2022*). While microfluidic chemotaxis assays have evolved since their inception (*Ahmed et al., 2010*), the vast landscape of potential chemical compounds, combinations of compounds, and concentration gradient conditions that regulate these important processes necessitates the development of new high-throughput devices.

Faced with a broad range of chemostimulant concentrations and gradients in their natural environment (*Stocker, 2012*), microorganisms, specifically prokaryotes, have evolved exquisite chemosensing abilities with variable degrees of specificity to nutrients, dissolved resources, toxins, and signaling molecules (*Adler, 1966*; *Berg and Brown, 1972*; *Stocker, 2012*). Some bacteria exhibit a dynamic sensing range spanning five orders of magnitude (*Lazova et al., 2011*; *Kalinin et al., 2009*; *Cremer et al., 2019*) and can detect nano-molar attractant concentrations (*Mao et al., 2003*), while marine invertebrate spermatozoa have a reported detection limit approaching the femto-molar scale (*Guerrero et al., 2010*). Quantifying the strength of chemotactic responses across varying concentration and concentration gradient conditions presents a key challenge to understanding microbial driven processes, extending far beyond their search for optimal metabolic activity conditions (*Keegstra et al., 2022*). For example, in marine microbial communities, the natural phycosphere surrounding individual cells (*Raina et al., 2022*) contains a diverse spectrum and concentration of metabolite and organic material (*Moran et al., 2022*), which are taken up by chemotaxing microbes (*Zimmerman et al., 2020*). Viral infection of microbes augments this process and is a principal mechanism (*Moran et al., 2022*; *Evans and Brussaard, 2012*) for transforming live biomass to readily available organic matter. Lysis (*Weinbauer et al., 2011*) and exudation (*Howard-Varona et al., 2022*) by virus infected cells releases a diverse range and concentration of metabolite and organic material (*Moran et al.,*

*2022*). Furthermore, chemotaxis is essential in initiating bacterial infections and pathogenicity for both animals and plants (*Matilla and Krell, 2018*). For example, in gastric infections pathogenic organisms rapidly colonize surfaces via chemotaxis, where a range of attractants from urea to amino acids and metals are presumed to enable localization and colonization on the host epithelium (*Keilberg and Ottemann, 2016*). Identifying the key metabolites and signaling chemicals which drive microbial chemotaxis necessitates new microfluidic tools capable of probing the wide scope and scale of chemotactic behaviors across a myriad of complex systems.

Microfluidic devices are widely accepted as an indispensable platform for targeted chemotaxis assays by enabling the quantification of both single cell and population-scale responses to precisely-controlled chemical gradients (*Ahmed et al., 2010*; *Li Jeon et al., 2002*). One class of chemotaxis microfluidic devices, termed stop-flow diffusion, relies on flowing a chemostimulant solution and buffer stream side-by-side in a microchannel. Upon halting the flow a slowly-evolving concentration gradient forms via diffusion (*Figure 1a and b*; *Seymour et al., 2010*; *Stocker et al., 2008*; *Mao et al., 2003*; *Ahmed et al., 2010*). Other devices generate steady chemical gradients by utilizing porous materials (*Ahmed et al., 2010*) or mimic diffusing marine hotspots using micro-well assays that entice and trap chemotactic microorganisms (*Lambert et al., 2017*; *Raina et al., 2022*). While these well-established assays accurately measure chemotactic motility in physically relevant concentration gradients (*Stocker et al., 2008*), they largely overlook the potential for high-throughput screening afforded by microfluidic devices. Recently, such high-throughput capabilities have been broadly showcased in other fields through the use of parallelized microfluidics for clinical testing of viruses (*Welch et al., 2022*), drug responses (*Sugiura et al., 2010*), and cell profiling (*Prakadan et al., 2017*). The development of an integrated microfluidic design - comprising parallelized chemotaxis assays on a single chip - would enable high-throughput characterization of microbial chemotactic responses. Relative to time-prohibitive conventional assays, rapid chemotaxis phenotyping could facilitate comparative studies and discoveries across different swimming microorganisms, chemostimulants, and concentration gradient conditions.

Here, we present a microfluidic multiplexed chemotaxis device (MCD) that enables high-throughput chemotaxis screening of swimming microorganisms to chemical stimuli across concentration gradient conditions that potentially span the microorganism's entire sensitivity range. The two-layer device architecture comprises a serial dilution layer that produces logarithmically diluted chemostimulant solutions (*Sugiura et al., 2010*) and a cell injection layer that introduces swimming cells, whilst minimizing both the footprint and operational complexity of the device (*Figure 1c–e*). On a single chip, the MCD simultaneously performs six stop-flow diffusion chemotaxis assays (including control), which span five orders of magnitude in chemostimulant concentration. The dilution, mixing, gradient generation, and flow performance are fully characterized (Materials and methods), and the MCD is validated against a conventional chemotaxis device for a known marine bacterial chemotaxis system (*Vibrio alginolyticus*). To demonstrate the device's efficiency, capabilities, and operational flexibility, the MCD is then used to rapidly quantify the chemotactic responses of different microbes to a variety of chemostimulants. Compared to existing microfluidic devices, the MCD enables chemotaxis studies with significantly higher throughput rates, and most importantly facilitates the simultaneous measurement of chemotactic responses across a range of concentration gradient conditions.

## Results
### Multiplexed microfluidic device as a platform for high throughput chemotaxis screening

To enable rapid and efficient chemotaxis screening of swimming microbes, we designed the multiplexed chemotaxis device (MCD) to perform six chemotaxis assays in parallel on a single microfluidic chip (*Figure 1*). The individual assays in the observation region are based on laminar flow patterning and established stop-flow diffusion methods (*Ahmed et al., 2010*; *Kirby, 2010*; *Stehnach et al., 2021*; *Stocker et al., 2008*; *Nguyen et al., 2019*), where rapid, parallel flow of chemostimulus (concentration, $C_i$) and buffer ($C = 0$) solutions maintain initially stratified fluid regions. Upon stopping the flow, a chemostimulus gradient forms via diffusion (*Figure 1a and b*). For each chemotaxis assay (*Figures 2 and 3*), a swimming cell solution is injected between the chemostimulus and buffer so their response may be observed and recorded. The MCD (*Figure 1c–i*) performs six simultaneous assays comprising

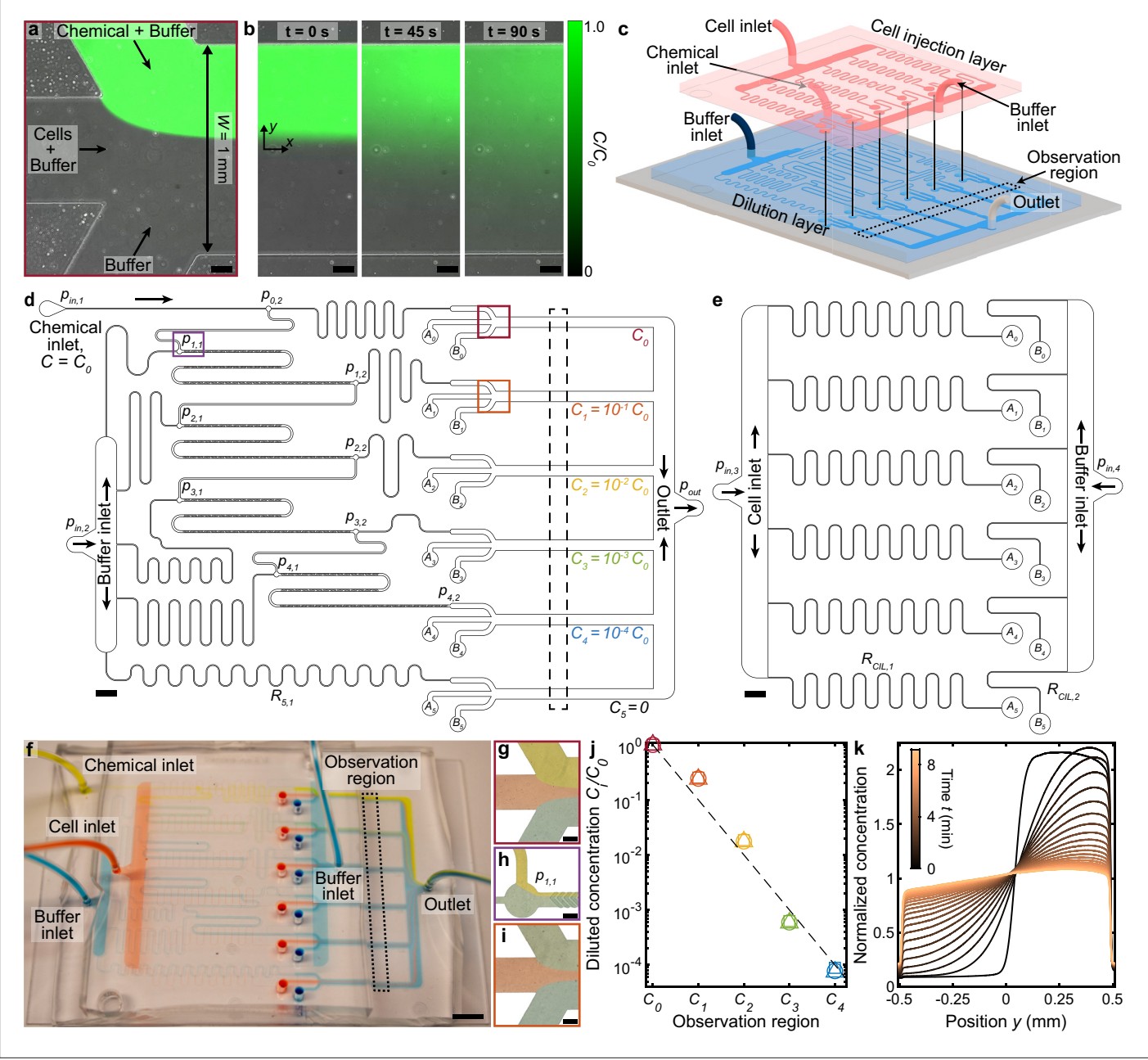

**Figure 1.** Multiplexed microfluidic device for simultaneous chemotaxis assays. (**a,b**) Continuous flow through a microfluidic junction (**a**) stratifies chemostimulus, cell, and buffer solutions, demonstrated here with fluorescein, DI water, and DI water, respectively. Upon halting the flow (**b**) diffusion establishes a chemical gradient across the channel, which is repeated at each observation channel in the MCD (d, red and orange boxes). Scale bars, 0.1 mm. (**c**) Assembly of the MCD showing the PDMS dilution layer (blue) and cell injection layer (red) microchannels mounted on a glass slide (grey; Materials and methods). (**d**) Scaled drawing of the dilution layer, which receives chemical (pressure, $p_{in,1}$) and buffer ($p_{in,2}$) solutions. Initial chemical concentration ($C_0$) is sequentially diluted 10-fold to each of four additional concentrations ($C_{1-4}$), plus a control solution ($C_5 = 0$). These six chemostimulus solutions are merged separately with additional cell ($A_i$) and buffer ($B_i$) solutions from the cell injection layer (**e**) for chemotaxis assays in respective observation channels (dashed black box, corresponding to c and f). (**e**) Scaled drawing of the cell injection layer which injects a cell suspension ($p_{in,3}$) and buffer solution ($p_{in,4} = p_{in,3}$) into the dilution layer ($A_i, B_i$; Materials and methods). Scale bars d,e, 2 mm. (**f**) Photograph of the completed MCD with dyed water to visualize the chemical (yellow), cell (red), and buffer (blue) fluid streams in the channel network. Scale bar, 5 mm. (**g**) Stratified chemical ($C_0$), cell, and buffer solutions in the first observation region (d, red box). (**h**) Dilution of the chemical ($C_0$) by the buffer prior to mixing in the first micromixer (**Stroock et al., 2002**) to produce concentration $C_1$ (d, purple box). (**i**), Stratified chemical solution after initial dilution ($C_1$, green) in the second observation region (d, orange box). Scale bars g-i, 0.2 mm. (**j**) Measured chemical concentrations (see Materials and methods) generated from the dilution microchannels (**d**) for various driving pressures $p_{in,1} = p_{in,2} = [100, 150, 200]$ mbar (square, circle, and triangle,

*Figure 1 continued on next page*

*Figure 1 continued*

respectively). (**k**) Measured evolution of the chemical gradient (**b**) produced in the $C_0$ observation region (*Figure 1—figure supplement 3*; Materials and methods) by the MCD shows the chemical diffusion across the channel with increasing time t.

The online version of this article includes the following figure supplement(s) for figure 1:

**Figure supplement 1.** Micromixer geometry and mixing performance.

**Figure supplement 2.** Hydraulic circuit design of MCD dilution layer and cell injection layer.

**Figure supplement 3.** Validation of chemostimulus distribution and gradient evolutionin in MCD observation regions.

**Figure supplement 4.** Two-layer photolithography and soft lithography microfabrication of the MCD.

five logarithmically decreasing chemical concentrations ($C_i = 10^{-i}C_0, i \in [0,4]$; *Figure 1j*), plus one control ($C_5 = 0$). The device is fabricated from polydimethylsiloxane (PDMS) in two layers (Materials and methods; *Figure 1c–e*). The primary function of the dilution layer (*Figure 1d*) is to receive two fluid inputs - base chemostimulus solution ($C_0$) and buffer ($C = 0$) - and passively generate six pre-defined concentration conditions ($C_i$) via a serial dilution process (*Li Jeon et al., 2002*; *Sugiura et al., 2010*; *Walker et al., 2007*), which are dispensed to each of the six observation channels for chemo-taxis assays. At each stage of the serial dilution process, the chemostimulus stream is combined with buffer in a 1:9 volumetric flow rate ratio, where efficient mixing of the solutions is necessary for accurate dilution and chemotaxis assays downstream. To ensure sufficiently mixed solutions, herringbone micromixers (*Stroock et al., 2002*; *Figure 1d and h* and *Figure 1—figure supplement 1*) were incorporated into each dilution stage. These structured microchannel surfaces generate a three-dimensional flow to induce chaotic mixing, and thus significantly reduce the mixing length (*Stroock et al., 2002*) and the overall footprint of the device (Materials and methods; *Figure 1—figure supplement 1*). To achieve the targeted chemostimulus concentrations in the dilution layer, the microfluidic network was designed using hydraulic circuit theory (Materials and methods; *Figure 1—figure supplement 2*), and the accuracy of the serial dilution process was experimentally confirmed (*Figure 1j*).

The cell injection layer (*Figure 1e*) introduces a suspension of swimming microbes and a sheathing buffer solution from two corresponding inlets into the six observation regions of the dilution layer (*Figure 1d*, dashed box). In the observation regions, the chemostimulus solution, cell suspension, and buffer streams comprise six standard stop-flow chemotaxis assays (*Figure 1a*), where each one incorporates a unique chemostimulus concentration. The observation channel width ($W = 1\,\text{mm}$) and height ($H = 90\,\mu\text{m}$) are similar to other microfluidic devices (*Stocker et al., 2008*; *Ahmed et al., 2010*; *Stehnach et al., 2021*) and ensure organisms with different sizes can be studied using the MCD. The initially steady flow rates stratify the three fluids in the observation region and localize the cells in a thin band in the channel center with equal width chemostimulus and buffer streams on either side (Materials and methods; *Figure 1g and i* and *Figure 1—figure supplement 3*). Upon halting the flow, a unique and highly reproducible chemical gradient is formed in each observation channel via diffusion, where the consistency of the transient concentration profiles across all observation channels were confirmed using fluorescence microscopy (Materials and methods; *Figure 1k*, *Figure 1—figure supplement 3*). The mixing effectiveness, fluid stratification, and gradient formation were validated over a range of applied pressures (approximately 100–200 mbar) and were found to be consistent across all six observation regions (*Figure 1—figure supplement 1* and *Figure 1—figure supplement 3*). This efficient two-layer architecture reduces the operational complexity of the device by decreasing the total number of fluid inlets (four) and reduces the footprint of the microfluidic chip.

Both layers of the MCD are fabricated from polydimethylsiloxane (PDMS) through standard soft lithography techniques (replica molding; Materials and methods). The final microfluidic chip is assembled by plasma bonding the dilution layer to a standard double-wide microscope slide (75 mm × 50 mm×1 mm) and subsequently aligning and bonding the cell injection layer on top (Materials and methods; *Figure 1c and f* and *Figure 1—figure supplement 4*). This reusable device (Materials and methods) is driven by a single pressure pump which maintains flow stratification (1–2 min) prior to each assay to ensure consistent initial gradient conditions for measuring cell responses. Pump and microscope automation enables the chemostimulus gradient to be reset for rapid replicate measurements. The design and operation of the MCD can accommodate most single-celled microorganisms, and efficient micromixer channels facilitate the use of a wide range of dissolved chemostimulants (*Figure 1—figure supplement 1*; *Stroock et al., 2002*). Due to variations in the replica mold fabrication process,

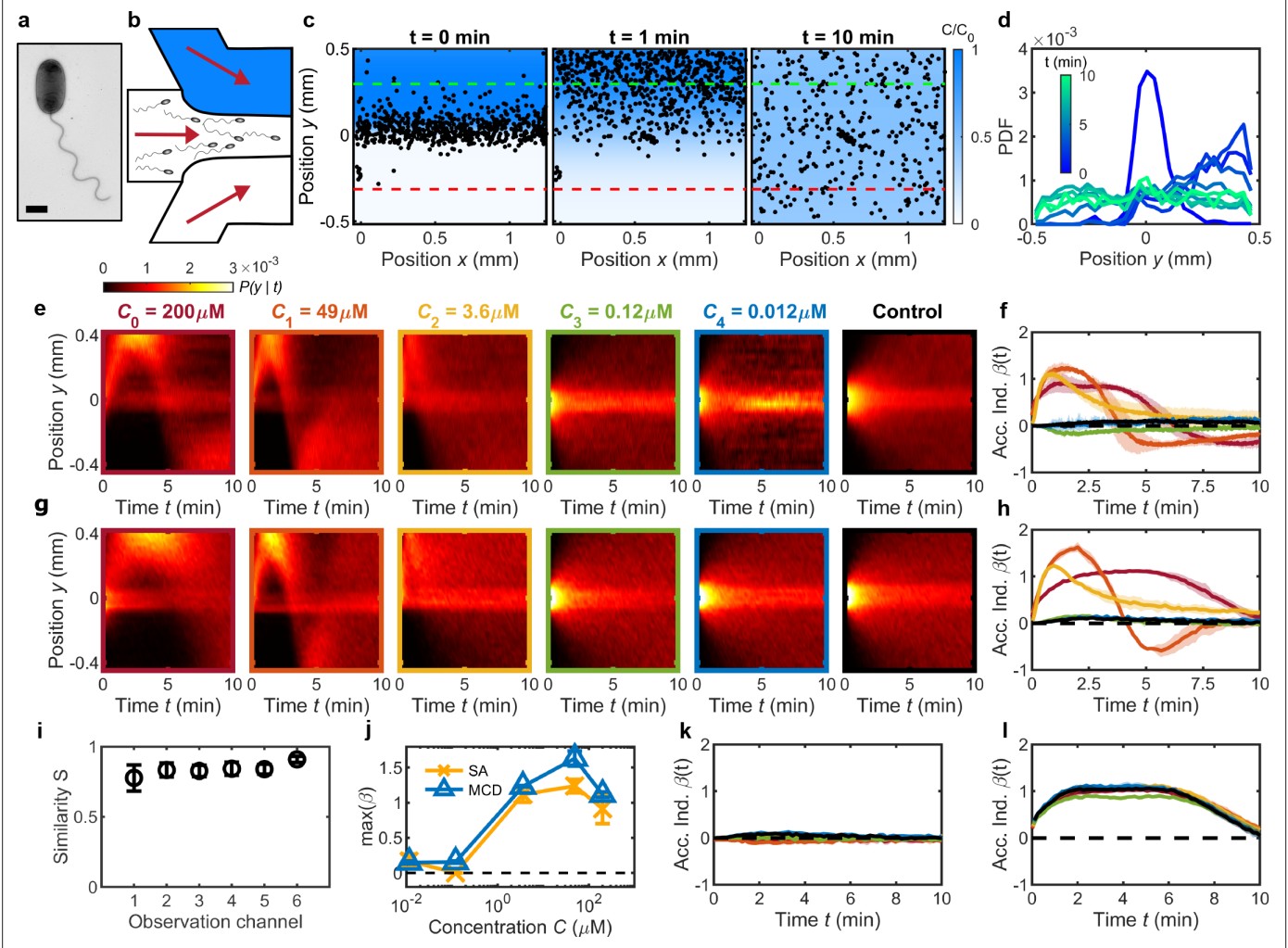

**Figure 2.** Validation of MCD and measurement of *V. alginolyticus* chemotactic performance toward serine. (**a**) TEM image of *V. alginolyticus* (Materials and methods). Scale bar, 1 μm. (**b**) A single chemotaxis assay (SA) with a single conventional microfluidic device flows chemostimulus (top, blue), cell suspension (middle), and buffer (bottom) streams into the observation region (Materials and methods). (**c**) SA with chemostimulus (serine, $C = 100\,\mu M$) showing measured cell positions (*V. alginolyticus*, black dots) at various times $t$ after initial flow stratification ($t = 0$ min) relative to the chemostimulus distribution (blue, from measurements in *Figure 1b*). Cells migrate up the gradient ($t = 1$ min) followed by uniform dispersal as the gradient dissipates ($t = 10$ min). Degree of cell accumulation is determined from the number of cells, $N_{p,n}$, in a 200 μm wide region on the chemostimulus side (positive; green dashed line) and buffer side (negative; red dashed line), respectively (*Seymour et al., 2010*; *Stocker et al., 2008*). (**d**) The measured cell distribution across the microchannel evolves over time (from c) and is represented as a conditional probability density of cell position, $P(y|t)$ (shown as a kymograph). (**e**) $P(y|t)$ for *V. alginolyticus* chemotactic response to serine from a series of SA devices having the same geometry as the MCD observation regions (*Figure 1*). SA measurements illustrate the transition from positive chemotactic response at high attractant concentration ($C_{0-2}$) to no response at low concentration ($C_{3-4}$) compared to control ($C_5 = 0\,\mu M$) (*Altindal et al., 2011*). (**f**) Accumulation index, $\beta(t)$, for SA measurements from e. (**g**) $P(y|t)$ measured by the MCD under the same conditions of the SA. (**h**) $\beta(t)$ measured from g accurately captures the behavior of *V. alginolyticus* to serine compared to SA results (**f**). (**i**) Sørensen similarity metric (*Cha, 2007*) comparing e and g, which is calculated at each time point and averaged. (**j**) Comparing MCD and SA peak chemotactic response quantified by max ($\beta(t)$) from f,h. (**k,l**) $\beta(t)$ in the absence of a chemical gradient (**k**; *Figure 2—figure supplement 1*) and for fixed gradients of $C_i = 200\,\mu M$ (**l**; *Figure 2—figure supplement 1b*) across each observation channel in the MCD indicates no significant bias. No gradient (**k**) conditions ($C_i = 0$) were obtained by injecting buffer into the chemical inlet (setting $C_0 = 0$). Fixed gradient (**l**) conditions ($C_i = 200\,\mu M$ of serine) were obtained by injecting $C_0 = 200\,\mu M$ of serine into both the chemical and buffer inlets of the dilution layer. Shading in f,h,k,l indicates one standard deviation (N=3). Error bars in i,j are one standard deviation across biological replicates.

The online version of this article includes the following figure supplement(s) for figure 2:

**Figure supplement 1.** Kymographs $P(y|t)$ for control chemotaxis experiments corresponding to *Figure 2k and l* with no chemostimulus present ($C_i = 0$) and with a fixed chemostimulus concentration ($C_i = 200\,\mu M$), respectively.

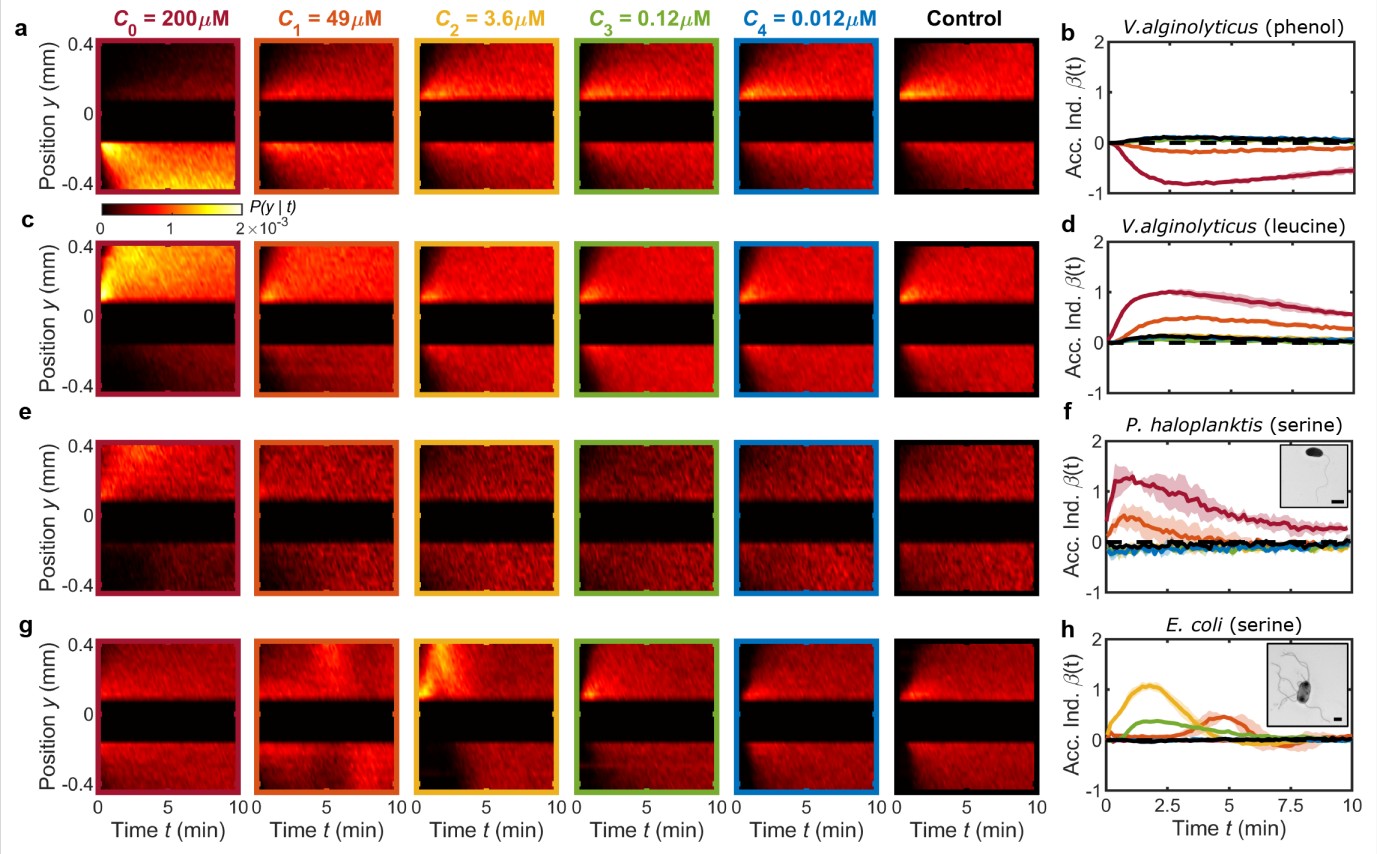

**Figure 3.** MCD enables rapid quantification of chemotactic responses across different chemostimulants and microbial species. (**a,b**) The negative chemotactic response of *V. alginolyticus* to the repellent phenol (*Homma et al., 1996*) is evident in kymographs of cell position, $P(y|t)$, and the accumulation index, $\beta(t)$, respectively. Central 250 μm wide band, which often contains a significant subpopulation of non-motile cells, is omitted from $P(y|t)$ for visualization purposes and has no impact on $\beta(t)$. (**c,d**) MCD measurements demonstrate the positive chemotactic response of *V. alginolyticus* to leucine observed in $P(y|t)$ and $\beta(t)$, respectively. (**e–h**) $P(y|t)$ in response to various concentrations of serine for bacteria *P. haloplanktis* (**e,f**) and *E. coli* (**g,h**). TEM images of *P. haloplanktis* (f, inset) and *E. coli* (h, inset). Scale bars, 1 μm. $\beta(t)$ for *P. haloplanktis* (**f**) illustrates a monotonically increasing response to increased concentrations (extracted from e). Accumulation index, $\beta(t)$, for *E. coli* (**h**) reveals a peak response to an intermediate serine concentration ($C_2 = 3.6\,\mu M$) and delayed accumulation at higher concentrations (*Bhattacharjee et al., 2021*). Shaded regions are standard deviation (N=2 and N=3 for *P. haloplanktis* and *E. coli*, respectively) across biological replicates. Color bar corresponds to kymographs in **a,c,e,g**.

The online version of this article includes the following figure supplement(s) for figure 3:

**Figure supplement 1.** Initial and final cell probability distributions ($P(y|t = 0, 10\,\mathrm{min})$) for the organism/chemostimulant systems in *Figure 3*.

a one-time tuning of the applied pressure ratio ($p_{in,1-2}/p_{in,3-4}$) for the dilution and cell injection layers is required to ensure symmetric flow in the observation channel (Materials and methods). Further optimization of the device layout could enable the number of observation channels to be expanded, including a broader range or more refined sampling of concentration gradient conditions. Additionally, the serial dilution layer design can be easily modified to produce different concentration scalings (e.g. logarithmic, linear; *Sugiura et al., 2010*; *Walker et al., 2007*). The high degree of parallelization for chemotaxis screening, combined with the demonstrated consistency and repeatability of the chemostimulus gradients, represents a significant advance relative to existing microfluidic chemotaxis devices (*Stocker et al., 2008*; *Ahmed et al., 2010*; *Li Jeon et al., 2002*).

## Validation of MCD performance against conventional chemotaxis assays

Having established the gradient generation performance, the MCD is compared to a conventional chemotaxis assay to (i) validate the chemotaxis measurements against a known microorganism-chemostimulus system and (ii) demonstrate the high-throughput capability of the MCD in comparison

with existing devices. The chemotactic response of the monotrichous marine bacterium *Vibrio algi-nolyticus* (*Figure 2a*) was measured using both the MCD and a single microfluidic gradient generation device (referred to as 'single assay', SA) (*Stehnach et al., 2021*). *V. alginolyticus* swims with a run-reverse-flick motility pattern, and it was chosen due to its well-documented chemotaxis (*Altindal et al., 2011*; *Son et al., 2013*; *Xie et al., 2011*) and prevalence within marine microbial communities (*Baker-Austin et al., 2018*). The SA device has an identical observation channel geometry with three inlets and operates using stop-flow diffusion in the same manner as the MCD, where the chemostimulus, cells, and buffer are initially flow-stratified (*Figure 2b*). Upon halting the flow for both assays, the concentration gradient develops via diffusion, and time-lapse microscopy is used to measure the evolution of the spatial cell distribution over time, $t$ (*Figure 2c*; Materials and methods). For example, in a gradient formed by the model chemoattractant serine (*Altindal et al., 2011*), cells initially confined to a central band migrate toward the attractant, before uniformly sampling the channel (*Figure 2c and d*) as the gradient dissipates within $t \approx 10\,\mathrm{min}$.

From one parallelized assay, the MCD precisely reproduces the chemotactic responses of *V. algi-nolyticus* toward various serine concentrations, compared to multiple, conventional single assays (*Figure 2e–h*). Chemotactic behavior of the bacteria was measured across a range of manually adjusted attractant concentrations for the single assay, decreasing from $C_0 = 200\,\mu\mathrm{M}$ and matching the serial dilution concentrations generated by the MCD (*Figure 1j*). For each concentration gradient, the spatial distribution (conditional probability) of cells, $P(y|t)$, across the observation channel, $y$, is shown over time as a kymograph (*Figure 2e and g*). For ease of comparison, the aggregate chemotactic response is distilled through the accumulation (or chemotactic) index, $\beta(t)$, which quantifies the relative fraction of cells responding to the chemostimulus (*Seymour et al., 2010*; *Stocker et al., 2008*). The accumulation index is defined as (*Seymour et al., 2010*; *Stocker et al., 2008*): $\beta(t) = (N_p(t) - N_n(t))/(N_p(T) + N_n(T))$, where $N_{p,n}(t)$ are the instantaneous number of cells at time t within predefined positive and negative accumulation regions (*Figure 2c*, green and red dashed lines, respectively), and normalized by the total number of cells in these two regions at the final time ($T \approx 10\,\mathrm{min}$). The measured cell distributions from the MCD (*Figure 2g*) were compared to the single assay results (*Figure 2e*) by calculating the Sørensen similarity metric (*Cha, 2007*) for each attractant concentration, showing excellent statistical agreement between the two assays (*Figure 2i*). A comparison of the strength of cell accumulation (*Figure 2j*) further emphasizes the high degree of concordance between the two devices. In particular, *V. alginolyticus* exhibits strong accumulation for high serine concentrations ($C_{0-2}$), above the previously reported chemotactic sensitivity threshold of 0.2 µM (*Altindal et al., 2011*). Consequently, no discernable response is observed for lower concentrations ($C_{3-4}$), which are comparable to $\beta(t)$ for the control ($C = 0$). At the highest concentrations ($C_{0-1}$) in both devices, the chemotactic motility exhibits a slight reversal at later times. This is most likely due to the relatively high concentration of *V. alginolyticus*, which rapidly consumes the available serine at the top of the channel. Combined with chemoattractant diffusion across the channel width, the concentration gradient flips direction at later times causing accumulation on the bottom side of the channel. The persistent central band of cells at later times is due to a sub-population of bacteria which remain non-motile over the course of each assay and are thus localised near $y = 0$ (*Figure 2e and g*; *Figure 3—figure supplement 1*). Because any non-motile or motile bacteria in this region do not impact the calculation of $\beta(t)$, the central band is omitted from future kymographs for visualization purposes. Finally, additional assays with no chemostimulus and with a fixed chemostimulus concentration confirmed the consistency of the chemotaxis assays across all observation regions (*Figure 2k and l* and *Figure 2—figure supplement 1*), which is expected from the verified gradient generation performance in each channel (*Figure 1—figure supplement 3*).

These validation assays serve to highlight the dramatically improved efficiency in chemical screening when compared to a standard single assay. In the single assay case, each chemical concentration requires: (i) manually diluting stock solutions, (ii) exchanging peripheral reservoirs for chemicals, and (iii) a new cell suspension for each concentration assay, all of which become extremely costly and time prohibitive, when considering the scope and scale of multi-chemical, -concentration, and -organism panel experiments. The single assay results (*Figure 2e and f*) required six different dilutions, cell solutions, and devices, and with three replicates per condition, required 18 individual assays. In contrast, the MCD collected the same data (*Figure 2g and h*) in only three automated assays, and did not require culture changes eliminating inconsistencies due to variations in growth media, dilution errors,

or growth conditions. If replacing the cell suspension is necessary, the MCD can be easily reset by simply exchanging the cell suspension with a fresh suspension and restarting the flow. Because it uses a robust serial dilution process and requires the preparation of a single chemostimulus ($C_0$), the MCD ensures consistent chemical concentrations across different experiments, a crucial factor when working with microorganisms having femto- to nanomolar chemotactic sensitivities (*Mao et al., 2003*; *Guerrero et al., 2010*; *Altindal et al., 2011*). The MCD fully screens both microbe and stimulus pairings with three replicates in ≈ 1 hour with a single cell culture, including the bench time associated with preparing the solutions (*Figure 2h*). In contrast, the panel of single assays (*Figure 2f*) requires nearly a full working day ($\approx 6 - 7\,\text{hours}$). Thus, the MCD is a powerful and much needed tool for large scale chemotactic panel studies, where consistency in stimulus concentration, elimination of biological variability, and the need for efficiency, are essential.

## Multiplexed microfluidic device supports high-throughput chemotaxis screening for novel stimuli and various microorganisms

Beyond validation with the single assay device, we demonstrate the efficacy of the MCD by examining the response of *V. alginolyticus* to both a known repellent and novel chemostimulus. Chemorepellents serve as an early warning sign for microorganisms to evade predators and toxins for survival (*Yang et al., 2015*). A single MCD assay reveals that *V. alginolyticus* exhibits negative chemotaxis ($\beta < 0$) to the model repellent phenol (*Homma et al., 1996*) with an observed detection threshold on the order of $C = 1 - 10\,\mu\text{M}$ (*Figure 3a and b*). Separately, the amino acid leucine has been identified as an important metabolite in human health (*Yang et al., 2020*) and marine environments (*Ferrer-González et al., 2021*; *Johnson et al., 2020*), and it serves as a measure of prokaryote heterotrophic activity from viral lysis in deep-sea environments (*Winter et al., 2018*). Previously reported as an attractant for marine prokaryotes (*Barbara and Mitchell, 2003*), we verify the positive chemotactic response of *V. alginolyticus* to leucine through rapid chemotaxis screening using the MCD (*Figure 3c and d*).

To further illustrate the capabilities and flexibility of the MCD, the chemotactic behavior of *Pseudoalteromonas haloplanktis* (*Figure 3f*, inset) and *Escherichia coli* (*Figure 3h*, inset) to serine was measured (*Figure 3e and g*) with no design changes to the MCD (see Materials and methods). *P. haloplanktis* is a rapid-swimming, monotrichous marine bacterium that is a model organism for chemotaxis to amino acids (*Barbara and Mitchell, 2003*) and exhibits strong chemotaxis towards cellular exudates (*Seymour et al., 2009*). Similar to *V. alginolyticus*, it utilizes a run-reverse-flick foraging strategy (*Son et al., 2016*) for efficient chemotaxis in patchy chemical landscapes (*Stocker et al., 2008*). *P. haloplanktis* exhibited a monotonically increasing chemotactic response (*Figure 3f*) with increasing serine concentration, qualitatively comparable to the response of *V. alginolyticus* to leucine (*Figure 3d*). In contrast to the marine prokaryotes, *E. coli* is a pathogenic bacterium, which uses the bundling and unbundling of its multiple flagella to perform run-tumble motility for migrating up or down chemical gradients (*Adler, 1966*; *Berg and Brown, 1972*). *E. coli* has served as the canonical organism for bacterial motility and chemotaxis (*Adler, 1966*; *Berg and Brown, 1972*; *Cremer et al., 2019*; *Mattingly et al., 2021*) and has been instrumental in our understanding of logarthimic-sensing (*Lazova et al., 2011*; *Kalinin et al., 2009*) and chemical navigation in complex physical environments (*Bhattacharjee et al., 2021*). A single MCD assay reveals that *E. coli* has a strong chemotactic response to intermediate serine concentrations ($C_2$; *Figure 3g and h*). The response significantly diminishes for high ($C_{0-1}$; *Figure 3g and h*) and low chemostimulant concentrations ($C_{3-4}$; *Figure 3g and h*). This observation reflects *E. coli*'s affinity for serine (*Yang et al., 2015*), but toxicity at high concentrations (*Neumann et al., 2014*). Furthermore, at higher concentrations ($C_1$), the initial accumulation is delayed in time (*Figure 3h*), a feature that is also consistent with the chemotactic sensitivity of *E. coli* to serine (*Son et al., 2016*; *Bhattacharjee et al., 2021*). Taken together, these three model swimming chemotactic microbes cover diverse foraging and motility strategies, whose range of chemotactic responses were efficiently screened using the MCD. These results demonstrate that the MCD can rapidly ascertain chemotactic responses across different chemostimulants and concentration ranges, which will be particularly valuable for studying the nano-molar and even femto-molar concentrations that characterize the detection thresholds of many microorganisms (*Mao et al., 2003*; *Guerrero et al., 2010*; *Altindal et al., 2011*).

The ability to simultaneously quantify a microbe's response for a spectrum of attractant concentrations using the MCD now enables rapid comparative studies across microbial or chemical species

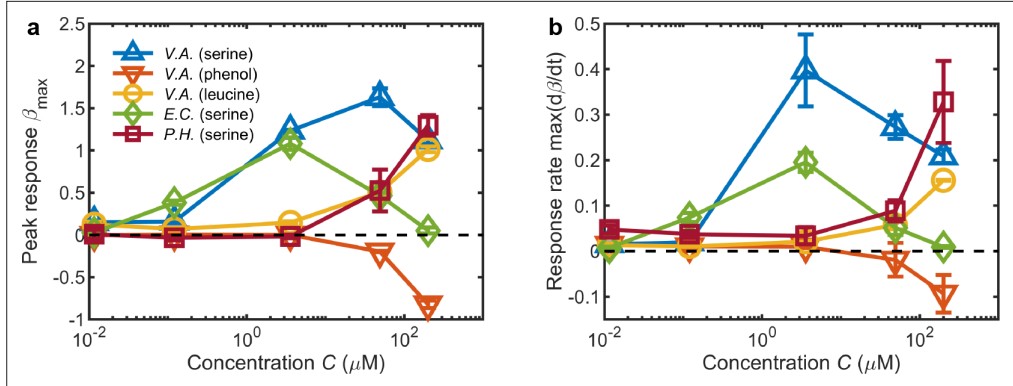

**Figure 4.** Summary of chemotactic responses across various species, chemostimulants, and concentrations measured using the MCD. (**a**) The primary metric for quantifying the chemotactic response of the bacteria was the peak of the accumulation index $\beta_{max} = \pm\max(|\beta(t)|)$, where the sign is determined by the positive or negative chemotactic behavior for each chemostimulus concentration (from *Figures 2h, 3b, d, f and h*). (**b**) The maximal response rate $\max(d\beta/dt)$ (prior to $\beta_{max}$) is indicative of the speed of cell accumulation. These metrics distinguish chemotactic behaviors, for example: The accumulation of *V. alginolyticus* to serine is greatest at high concentrations (**a**; $C = 49\,\mu M$), but the fastest response occurs at weaker concentrations (**a**; $C = 3.6\,\mu M$). Error bars are one standard deviation across biological replicates.

(*Figure 4a*). The magnitude of a microbe's response to a given concentration gradient is compactly summarized by the maximum (or minimum for negative chemotaxis) of their accumulation index, $\beta_{max} = \pm\max(|\beta(t)|)$, where the sign of $\beta_{max}$ is determined by the sign of the chemotaxis. In the case of serine, *V. alginolyticus* and *E. coli* (*Figure 4a*) both appear to have developed a chemotactic affinity to an optimal concentration (*Lazova et al., 2011*; *Bhattacharjee et al., 2021*), which is evident from the local maxima of $\beta_{max}$ and is intrinsically linked to their motility and sensing abilities. This analysis is complemented by examining the maximal response rate $\max(d\beta/dt)$ (measured prior to $\beta_{max}$; *Figure 4b*). The peak accumulation of *V. alginolyticus* to serine occurred at $C_1 = 49\,\mu M$ while the fastest rate of response occurred at a lower chemostimulus concentration ($C_2 = 3.6\,\mu M$). Despite its similar morphology and motility to *V. alginolyticus*, *P. haloplanktis* (*Figure 4a*) exhibits a monotonically increasing response to serine across the concentrations tested, with the fastest response rate also occurring at the highest concentration tested (*Figure 4b*; $C = 200\,\mu M$). Separately, the lack of an optimal concentration in the response of *V. alginolyticus* to leucine likely indicates a higher saturation threshold in the relevant chemoreceptors, relative to its serine response. This demonstrated screening efficiency highlights the benefits of this new microfluidic platform for tackling large-scale chemotaxis studies in a complementary manner to existing tools (*Lambert et al., 2017*; *Raina et al., 2022*; *Haring et al., 2020*; *Satti et al., 2020*), for a diverse array of micro-organisms and compounds.

## Discussion

Here, we have introduced a novel microfluidic multiplexed chemotaxis device for rapid quantification of bacterial responses to a range of chemostimulant concentrations. Identifying the diverse chemical compounds and concentrations responsible for driving microbial interactions that underpin important environmental and human health processes – for example, ecosystem scale nutrient cycling and disease transmission – has proven to be a tedious and monumental undertaking. A primary challenge is the sheer diversity of potential compounds and the extensive concentration range of microbial responses from micro- to femtomolar (*Mao et al., 2003*; *Guerrero et al., 2010*; *Altindal et al., 2011*). Existing chemotaxis assays, whilst able to screen multiple compounds, are ill-equipped to probe the temporal response of motile cells and are limited to low time-resolution (e.g. hours; *Lambert et al., 2017*). The multiplexed chemotaxis device (MCD) design presented here mitigates these restrictions by rapidly screening the chemotactic behavior of microbes across a spectrum of chemostimulus conditions simultaneously (*Figure 1f*) on significantly faster timescales. This work opens new avenues to large-scale, laboratory-based panel experiments previously inaccessible with existing microfluidic devices.

The MCD's two-layer device architecture uses a single pressure source to serially dilute a chemo-stimulus with a single buffer input, producing five logarithmically separated chemical solutions (*Figure 1j*). Separately, a cell suspension and additional buffer stream are introduced into each of the six observation regions, where the chemotactic response of the cells is recorded. The MCD performs reliably for a wide range of applied pressures and only requires calibration once, provided that the PDMS microchannels are cast from the same molds (Materials and methods). The simultaneous chemotaxis assays are not only fast and efficient compared to conventional (single assay) devices, but they also eliminate variability due to multiple culture preparations and potential changes in cell cultures that may occur over time (*Figure 2g*). Taken together, our results illustrate that the MCD enables robust and efficient quantification of the chemotactic responses of various bacterial species to different chemostimulants, simplifying the labor-intensive chemotaxis screening process (*Figure 4*).

The MCD design is amenable to a range of modifications to tailor its application, including but not limited to, alteration of chemical concentrations, gradients and flows, and increased multi-plexing. For example, the design could be altered to enable the retrieval of chemotaxing cells from the individual observation regions for additional downstream analysis (*Mao et al., 2003*; *Takagi et al., 2005*; *Gurung et al., 2020*; *Konishi et al., 2020*). The sensitivity of prokaryotes is intrinsically linked to the strength of the concentration gradient (*Lazova et al., 2011*; *Mao et al., 2003*; *Kalinin et al., 2009*; *Mattingly et al., 2021*), an effect which can be examined with the MCD. With no alterations to the design or operation, simply changing the concentration of the input chemostimulus ($C_0$, *Figure 1d*) will shift the examined concentration range. Likewise, the background concentration across all observation channels is adjustable by exchanging the buffer solution input in the cell injection layer for a chemostimulus solution. The serial dilution layer hydraulic circuit design can be modified to produce specific dilution ratios (i.e. linear or logarithmic; *Sugiura et al., 2010*; *Walker et al., 2007*), where optimization of the device layout can expand the number of different chemical conditions probed. The device operation is demonstrated here using fast swimming bacteria. However, the MCD can be operated using steady flow (*Ahmed et al., 2010*; *Li Jeon et al., 2002*) to study chemotaxis in slow-moving, surface attached microorganisms in systems such as neutrophil chemotaxis (*Li Jeon et al., 2002*) and biofilm formation (*Boyeldieu et al., 2020*). While the work presented here focuses solely on prokaryotes, the current device geometry will accommodate larger eukaryotic cells ($\approx 10\,\mu$m) and in principle could be scaled up for larger multicellular microorganisms (*Varennes et al., 2017*).

Whilst the MCD does offer substantial improvement and throughput compared to existing alternatives, it is not without limitations. Typical microfluidic devices consist of a single layer and can be fabricated in most cleanrooms, but the multi-layer dilution layer (*Figure 1—figure supplement 4*) does require the precision of a mask aligner (Materials and methods). If such facilities are not available, microchannel molds can be fabricated using other techniques such as 3D printing (*Su et al., 2023*) and micromilling (*Guckenberger et al., 2015*). A two-layer PDMS approach was chosen here to reduce the overall footprint of the device, but care must be taken to align the two layers, as misalignment could render the device inoperable, for example due to misconnecting ports between layers. In the current mode of operation, each observation channel is only imaged once every approximately 8 s, which is usually insufficient to identify swimming speed changes or changes in motility patterns associated with chemotaxis. This issue could be mitigated by capturing a short video at each visit to the observation channels, but it will also decrease the overall sampling frequency across concentration conditions.

In summary, the MCD provides a robust, high-throughput tool for expediting microbial chemotaxis screening. This engineered microfluidic device could simplify the study of microbial chemotaxis, which is paramount to understanding and modeling diverse problems including global scale carbon and nutrient cycling (*Moran et al., 2022*). Such technology could also be applied to accelerate microfluidic approaches to human health studies (*Song et al., 2018*; *Atmaramani et al., 2019*; *Harimoto et al., 2022*; *Chen et al., 2020*). Microfluidics have been identified as a potential means for meeting the high-throughput demands of chemical synthesis, screening, and testing with living cells, applications that remain key issues in drug discovery (*Dittrich and Manz, 2006*) and in meeting the challenge of antibiotic-resistant microbes (*Qin et al., 2021*).

## Materials and methods
### Microfluidic device design
Hydraulic circuit framework

In analogy with electrical circuits, well-established hydraulic circuit theory (*Oh et al., 2012*) was used to design the complex microfluidic network of the multiplexed chemotaxis device (MCD; see *Figure 1* and *Figure 1—figure supplement 2*). Briefly, for incompressible, laminar flow through a constant cross-section microchannel, the pressure drop, $\Delta p$, is linearly proportional to the volumetric flow rate, $Q$, and is given by the Hagen-Poiseuille law (*Oh et al., 2012*; *Kirby, 2010*): $\Delta p = QR$. The hydraulic resistance, $R$, is a function of the fluid viscosity (properties of water assumed for all fluids) and the channel geometry. Fabrication of microfluidic devices via the soft lithography method (*McDonald et al., 2000*) used here (see below) results in rectangular cross section microchannels (height, $H$; width, $W$; length, $L$). Exact expressions for $R$ are tabulated for rectangular and other cross-section channels and provided in various resources (*Pozrikidis, 2001*). Combined with conservation of mass, $\Sigma Q_i = 0$ at the junctions (nodes) between several channels, $i$, the Hagen-Poiseuille law enables us to design complex microfluidic networks (*Figure 1—figure supplement 2*) via the solution of a set of linear equations.

MCD design considerations

The primary goal of the MCD was to efficiently perform several stop-flow bacterial chemotaxis assays (*Ahmed et al., 2010*; *Ford et al., 1991*) simultaneously for a range of chemostimulus concentrations. The design requirements were to: (i) dilute and distribute five logarithmically spaced concentrations of chemostimulus plus one control buffer solution to each of six chemotaxis assays. (ii) Perform those six chemotaxis assays in parallel on the same microfluidic chip. And, (iii) the microfluidic device should receive minimal fluid inputs to reduce setup time. The MCD has a two-layer architecture (dilution layer and cell injection layer) with a total of four fluid inputs and one (waste) output (*Figure 1* and *Figure 1—figure supplement 2*). Each having two inputs, the dilution layer and cell injection layer are designed to be regulated by a pressure-driven flow controller operating at a pressure, $p_{in,1-2} \approx 100\,\mathrm{mbar}$ and $p_{in,3-4} \approx 50\,\mathrm{mbar}$, respectively, while the lone output is at atmospheric pressure ($p_{out} = 0$). The dilution layer receives a base concentration chemostimulus solution (concentration, $C_0$; $p_{in,1}$) and a buffer solution ($C = 0$; $p_{in,2}$), and the cell injection layer receives a bacterial suspension in buffer ($p_{in,3}$) and a second buffer solution ($C = 0$; $p_{in,4}$). The serial dilution (*Sugiura et al., 2010*) process sequentially combines the chemostimulus and buffer fluids to produce separate microchannel streams having chemostimulus concentrations of $C_i = 10^{-i}C_0$ ($i \in [0, 4]$) and $C_5 = 0$ (control). The resulting six diluted chemostimulus solutions are merged in separate observation channels with a flow-stratified bacterial suspension and second buffer stream, which eventually forms the chemostimulus gradient in the chemotaxis assay. The three solutions are designed to symmetrically occupy the following fractional widths of the observation channel (total width, $W$): $w_1 = 4W/9, w_2 = W/9$, and $w_3 = 4W/9$. The observation channel width ($W = 1\,\mathrm{mm}$) and height ($H = 90\,\mu\mathrm{m}$) were chosen to set the chemostimulus gradient strength based on a physically relevant length scale (*Ahmed et al., 2010*; *Stehnach et al., 2021*; *Mattingly et al., 2021*) and to ensure that the upper and lower microchannel walls do not impede bacterial motility, respectively. These dimensions are in line with conventional microfluidic chemotaxis assays (*Ahmed et al., 2010*), and as a consequence of the microfabrication process, the chosen $H$ sets the height of the dilution layer channels excluding the micromixer. The initial flow rate of the three streams in the observation region (*Figure 1d*) was designed to be $20, 5, 20\,\mu\mathrm{l}\,\mathrm{min}^{-1}$ for the chemostimulus ($Q_{out}$), cell ($Q_{CIL,1}$), and buffer solution ($Q_{CIL,2}$), respectively (*Figure 1—figure supplement 2*). These flow rates maintain the stratification of the cell suspension with a 4:1:4 ratio, while ensuring that the bacterial cells are not damaged by the flow (*Figure 1—figure supplement 3*). Beyond these design requirements, the geometries - and thus hydraulic resistances – of several components are set independently, including: micromixer channels $R_M$, bridge channels $R_B$, and observation channels $R_{4,4}$ (*Figure 1—figure supplement 2* and Appendix 1). Based on these design requirements, hydraulic circuit theory was used to determine the required resistances of each microchannel in the MCD network, and subsequently the microchannel geometries (*Oh et al., 2012*). A complete list of the microchannel resistances and dimensions for the final design is provided in Appendix 1.

## Herringbone micromixer design

### Mixing performance

For the serial dilution process to perform as designed, effective mixing of the chemical solution and buffer are critical. Here, we use a well-established herringbone micromixer geometry (*Stroock et al., 2002*), where a series of ridges on the upper wall of an otherwise rectangular microchannel (*Figure 1—figure supplement 1a and b*) drive a three-dimensional flow to enhance mixing (*Stroock et al., 2002*; *Ottino and Wiggins, 2004*). A separate microchannel - having the same cross section geometry as the MCD design - was fabricated to independently quantify mixing performance and to select the necessary mixer length. The test micromixer channel was 41.3 mm long with 29 mixing cycles (comprised of two sets of six alternating herringbone ridges each). Two aqueous solutions of fluorescein salt (Sigma; concentrations, $C_{50} = 50\,\mu$M and $C_{10} = 10\,\mu$M) (*Petrášek and Schwille, 2008*) were injected individually into the MCD, and calibration images of dye intensity were captured after each herringbone mixer cycle (*Figure 1—figure supplement 1*), corresponding to the maximum ($\mathbf{I}_{50}$) and minimum ($\mathbf{I}_{10}$) dye concentrations, respectively. The region within 20 μm of the microchannel walls was excluded from analysis due to reflection and refraction effects (*Stroock et al., 2002*). Subsequently, the two solutions were flowed side-by-side with images ($\mathbf{I}_i$) recorded in the same locations as above and normalized as follows:

$$\mathbf{I} = \frac{\mathbf{I}_i - \langle \mathbf{I}_{10} \rangle}{\langle \mathbf{I}_{50} \rangle - \langle \mathbf{I}_{10} \rangle},$$

where $\langle \cdot \rangle$ denote spatial averaging. The degree of mixing is defined as (*Stroock et al., 2002*), $\mathrm{DOM} = \sqrt{\langle (\mathbf{I} - \langle \mathbf{I} \rangle)^2 \rangle}$, where values of 0.5 and 0 indicate fully non-mixed and mixed solutions, respectively. This measurement (*Figure 1—figure supplement 1c*) was repeated for both the designed flow rate for the MCD ($Q_M = 22\,\mu l\,\mathrm{min}^{-1}$) and for a second higher flow rate ($222\,\mu l\,\mathrm{min}^{-1}$). Based on standard metrics (*Stroock et al., 2002*), the two solutions are considered mixed when $\mathrm{DOM} \leq 0.05$ (i.e. 90% complete mixing). For both flow rates, this criterion is met after 9 complete herringbone ridge cycles, and a final design with 26 herringbone cycles was chosen for the MCD. The independence of mixing efficiency on flow rate, combined with a safety factor of approximately three for the number of herringbone cycles, ensures that the serial dilution portion of the MCD will perform accurately for a wide range of chemostimulants and flow speeds.

### Micromixer hydraulic resistance

To complete the design of the MCD, it was necessary to determine the hydraulic resistance of the herringbone micromixer $R_M$ which was measured empirically using a parallel microfluidic device (*Choi et al., 2010*). Briefly, a microfluidic device was fabricated with two parallel channels connected by shared inlets and outlets. The parallel channels had identical rectangular geometries except one had the herringbone ridges replicating the micromixer channel section $R_M$ (*Figure 1—figure supplement 4*). Two solutions, DI water and $1\,\mu l\,\mathrm{ml}^{-1}$ tracer particle suspension (0.25 μm radius; 2% solid; carboxylated FluoroSpheres, Life Technologies), were flowed through the device using glass syringes (2.5 ml; Hamilton) mounted on two separate syringe pumps (Harvard Apparatus). The particle solution was visualized using fluorescence microscopy, and the flow rates of the two pumps were adjusted such that the two streams divided equally into the parallel channels. The micromixer hydraulic resistance was determined from the resulting flow rate ratio and the known (analytical) resistance of the non-mixer channel (*Oh et al., 2012*; $R_M = 0.0043\,\mathrm{mPa\,s\,\mu m}^{-3}$; *Figure 1—figure supplements 2 and 4c*), and the results were corroborated by COMSOL Multiphysics simulations (not shown; *Stehnach, 2022*).

## Microfabrication and assembly

Microfluidic channel molds were fabricated using standard single and two-layer photolithography (*Anderson et al., 2000*) to transfer the final channel designs from a photomask (Artnet Pro, formally CAD/Art Services, Inc) onto a silicon wafer (100 mm diameter; University Wafer), which was spin-coated with photoresist (SU-8; Kayaku Advanced Materials). The single assay chemotaxis devices and MCD cell injection layer were made using SU-8 2050 and 2025, respectively, and multilayer devices (micromixer validation channels, MCD dilution layer) were made using SU8-3050 and SU8-2025 for the main rectangular channels and herringbone ridges, respectively. The ridges of the micromixers

(*Stroock et al., 2002*) were applied by halting the first-layer photolithography after the first post-exposure bake (PEB), spin-coating a second layer of SU-8 photoresist onto the wafer, then completing the remainder of the photolithography processes as usual (*Anderson et al., 2000*). The ridges of the herringbone micromixer (*Stroock et al., 2002*) extend over the main channel by $\approx 10\,\mu$m on both sides to account for potential misalignment during the multilayer photolithography (*Figure 1—figure supplements 1–4*). The final channel heights for the fabricated MCD dilution layer (*Figure 1d* and *Figure 1—figure supplement 4a–c*) were $90 - 94.5\,\mu$m, $37 - 38.5\,\mu$m for the main channel and herringbone ridges, respectively, while the cell injection layer (*Figure 1e*) was $71 - 73\,\mu$m high (Bruker's DekTak).

The MCD was fabricated using two-layer soft lithography (*McDonald et al., 2000*) with polydimethylsiloxane (PDMS; Sylgard 184) at a 10:1 (elastomer:curing agent) ratio. All channel wells were punched using a 1.5 mm diameter biopsy punch (Integra). The dilution layer mold was first silanized through vapor deposition (*Sidorova et al., 2009*) in a vacuum desiccator with 1–2 drops of tridecafluoro-1,1,2,2-tetrahydrooctyl trichlorosilane (Gelest Inc) to help release the cast PDMS. Post-silanization, PDMS was poured onto both the cell injection layer and dilution layer molds and degassed in a vacuum chamber ($\approx 1$ hour) prior to curing (65°C for $\approx 1$ hour). The resulting PDMS dilution layer channel was first plasma bonded onto a standard thickness, double wide glass slide (75 mm × 50 mm×1 mm; Fisherbrand) using a plasma oven (Plasma Etch Inc), and subsequently heated on a hot plate at 110°C for one hour to promote covalent bonding (*McDonald et al., 2000*; *Figure 1—figure supplement 4d*). Next, the cell injection layer was plasma bonded on top of the dilution layer, with care taken to ensure the alignment of the fluid wells connecting the two layers (*Figure 1—figure supplement 4e*). The assembled device was baked again on a hot plate at 110°c for 1 hr. The PDMS-PDMS bond was found to be sufficiently strong for the relatively low pressure applications of the MCD (*Eddings et al., 2008*). Before injecting any fluids into the MCD, the microchannels should be inspected to insure no debris is blocking a channel. Debris (e.g. dust introduced during the fabrication process) that clogs or partially clogs the microchannels could negatively impact performance by changing the hydraulic resistances of the individual channel (*Figure 1—figure supplement 2* and Appendix 1). Particulates can potentially be removed by flushing the device, but as with many microfluidic applications, large obstructions may render the device unusable. All other devices (e.g. single assay chemotaxis devices and micromixer validation channels) were fabricated using single-layer soft lithography, where an individual PDMS device was molded and subsequently bonded to a standard microscope slide using the procedures described above.

## MCD dilution, flow, and gradient generation performance

The performance of the fabricated MCD was validated using epifluorescence microscopy (Nikon Ti-E) with an aqueous fluorescent dye (fluorescein sodium salt, Sigma) in various concentrations (described below). Images of the dye distribution were captured at the midplane of the channels with a sCMOS camera (Zyla 5.5; Andor Technology). Fluorescein was chosen due to its similar diffusion coefficient with the chemostimulant serine (*Altindal et al., 2011*). Minor deviations in the performance of the MCD from the original circuit design (*Figure 1—figure supplement 1*) are likely due to variations in the fabricated channel mold heights (Appendix 1). Such variations impact the hydraulic resistances (*Oh et al., 2012*) and symmetry of the cell solution (*Figure 1—figure supplement 3*).

### Serial dilution

The primary function of the MCD dilution layer is to sequentially dilute the input chemical solution ($C_0$) with buffer to generate four logarithmically decreasing concentrations ($C_{1-4}$) for each of the observation channels (plus one control, $C_5 = 0$). The dilution performance was quantified by injecting a solution with known fluorescein concentration ($C_0 = 1$ mM). The diluted concentration field, $C_i(x, y)$, at each of the observation channels was then determined from the local measured dye intensity, $\mathbf{I}_i(x, y)$, which are linearly proportional, $C_i(x, y) \propto \mathbf{I}_i(x, y)$. Fluorescence images were recorded upstream of the inlet before the three fluids made contact in each observation channel (*Figure 1a*) using a 20× (0.45 NA) objective. Due to the strong 10-fold dilution, pairs of images were acquired for adjacent channels with optimized exposure times to account for the finite dynamic range of the camera. The mean normalized concentrations provided for each observation region from the serial dilution process were reconstituted from the measured image intensity as follows:

$$\frac{C_i}{C_0} = \prod_{n=1}^{i} \frac{\langle \mathbf{I}_n(x, y) \rangle}{\langle \mathbf{I}_{n-1}(x, y) \rangle}, \quad n \in [1, 4],$$

where the angled brackets indicate the spatial average. The resulting serial dilution followed the expected logarithmic (10-fold) dilution $C_i/C_0 = 10^{-i}$ for $i \in [0, 4]$ for which the system was designed (*Figure 1j*). This measurement was performed for three different sets of applied driving pressures, which yielded nearly identical results and illustrated the robustness of the serial dilution process.

## Stratification symmetry

The symmetry of the stratified chemostimulus and buffer distributions in the observation channel is critical to prevent bias in the chemotaxis measurements. As minor errors in the manufacturing process can alter this symmetry, the applied pressure for the cell injection layer ($p_{in}, 3, 4$; *Figure 1—figure supplement 2*) was tuned until the widths of the chemical, cell, and buffer streams in each observation channel were 4:1:4 ratio, respectively. Tuning was visualized by flowing a fluorescein solution ($C_0 = 100\,\mu\text{M}$) in both the chemical and buffer inlets of the dilution layer as well as the buffer inlet of the cell injection layer (*Figure 1d and e*). The ratio of applied pressures ($p_{in,1-2}/p_{in,3-4}$) between the dilution and cell injection layer remained the same for all chemotaxis assays ($p_{in,1-2}/p_{in,3-4} = 10/7$), which was slightly lower than the designed value ($p_{in,1-2}/p_{in,3-4} = 2$). Tuning is only required for the first device fabricated from a particular set of molds, after which the calibration and tuning applies to all subsequent devices fabricated from the same mold set due to the robust nature of soft lithography. If significant variations in the stratification symmetry occur in the observation regions, the MCD will not function properly, where possible causes include: (*i*) mis-alignment of the two PDMS layers and/or (*ii*) debris blocking or impeding the flow. In the latter case, if inspection of each microchannel and flushing (see: Experiment replicates and device reusability) is ineffective, a new device is recommended.

## Chemostimulus gradient consistency

Beyond ensuring the symmetry of the chemostimulus and buffer stratification, the time evolution of the resultant chemostimulus gradient must be consistent across each of the observation channels to accurately compare bacterial chemotactic responses. The chemical gradient evolution (*Figure 1k* and *Figure 1—figure supplement 3c and d*) was measured by first flowing a fluorescein solution ($C_0 = 100\,\mu\text{M}$) through both the chemical and buffer inlets of the dilution layer and DI water through both inlets of the cell injection layer. Having independently verified the performance of the serial dilution process, this approach produces identical base concentrations for all observation channels, $C_i = C_0$, and thus, enables easy comparison of the resulting concentration gradients in each channel. Upon halting the flow, an image was recorded (10×, 0.3 NA objective) in each observation channel every 5 s for approximately 9 min. The time evolution of the (normalized) spatial fluorescence intensity was measured to visualize the chemical gradient. The resulting concentration profiles were found to be highly consistent across the various observation regions and for different driving pressures (*Figure 1—figure supplement 3c and d*).

## Cell culturing

*Vibrio alginolyticus* (YM4; wild-type) from -80°C stock were grown overnight in Marine 2216 media (Difco) by incubating at 30°C and shaking at 600 revolutions per minute (RPM). The overnight culture was diluted 100-fold into fresh pre-warmed 2216 media and grown for three hours (30°C, shaking at 600 RPM) to O.D. $\approx 0.2$. 7 ml of culture was then washed and resuspended (1,500 RCF for 5 min) in 4 ml of artificial seawater (ASW).

*Psuedoalteromonas haloplanktis* (ATCC 700530) from -80°C stock were grown overnight in Marine 2216 (Difco) media by incubating at room temperature and shaking at 100 RPM (*Stocker et al., 2008*).

*Escherichia coli* (MG1655) from -80°C LB stock were grown overnight in Terrific Broth (TB, Sigma Aldrich) by incubating at 34°C and shaking at 220 RPM (*Stocker et al., 2008*). The overnight culture was diluted 100-fold into fresh pre-warmed TB media, and grown for approximately three hours (34°C, shaking at 220 RPM) to O.D. $\approx 0.5$. 8 ml of culture was then washed three times and resuspended (4000 RCF for 5 min) in 4 ml of motility buffer (10 mM potassium phosphate, 0.1 mM EDTA, 10 mM NaCl, pH 7, filter sterilized 0.2μm). A total of 16 ml of culture was washed twice and resuspended (1200 RCF for 5 min) in 6 ml of ASW.

## Media and chemostimulants

Artificial seawater (ASW) was prepared following the NCMA ESAW Medium recipe, which was adapted from *Harrison et al., 1980* and modified (*Berges et al., 2001*). ASW was used as the buffer and the chemical solvent for chemotaxis assays for both *V. alginolyticus* and *P. haloplanktis*, while motility buffer (see above) was used for chemotaxis assays with *E. coli*. Chemostimulus materials were purchased from Sigma Aldrich for use in the chemotaxis experiments, specifically: serine (S4500), phenol (P1037), and leucine (L7875).

## Microfluidic chemotaxis assays

Prior to use, the MCD was pre-treated by flowing a 0.5% (w/v) bovine serum albumen solution (BSA; Sigma Aldrich) to reduce cell adhesion to the microchannel surfaces. The device was flushed for over 10 min prior to first use with the cell, chemostimulus, and buffer suspensions. For chemotaxis assays, fluid flow was driven by a single pressure controller (Elveflow OB1; 1 mbar =100 Pa): $p_{in,1-2} = 200\,\text{mbar}$ (dilution layer) and $p_{in,3-4} = 140\,\text{mbar}$ (cell injection layer). Pressures were scaled down to 100 mbar and 70 mbar, respectively, for *P. haloplanktis* experiments. Between each chemotaxis assay, the fluid inputs were flowed for a minimum of 2 min to stratify cell, chemostimulus, and buffer streams in the observation channels. Upon halting the flow, a monotonic concentration profile was established in each observation channel due to the diffusion of the chemostimulus (*Figure 1* and *Figure 1— figure supplement 3*). The spatio-temporal evolution of the bacterial distribution was determined by imaging the cells using phase-contrast microscopy (10×, 0.3 NA objective; Nikon Ti-E) with a sCMOS camera (Zyla 5.5, Andor Technology) for approximately 10 min. An automated computer-controlled stage was used to cyclically move the microscope field of view to each observation channel 75 times, producing an effective imaging period of 8 s for each observation channel. Each experiment was technically replicated at least three times with the same culture and repeated on different days with freshly grown cells. For validation of the MCD, a conventional single assay (SA) microfluidic device (*Figure 2b–f*) was designed with a similar geometry to the individual MCD observation channels. Specifically, the SA devices had three inlets (width, 0.5 mm) which merged in a single observation channel. The devices were fabricated in a single layer using soft lithography (see above), and they were pre-treated with a 0.5% (w/v) BSA solution and flushed with ASW prior to experiments. The SA chemotaxis devices were used to validate the MCD for the well-established chemotactic behavior of *V. alginolyticus* to the chemoattractant serine (Sigma). The three inlets of the single assay device (*Figure 2b*) carried the chemoattractant dissolved in ASW, *V. alginolyticus* suspended in ASW, and ASW, respectively. The three solutions were flow stratified for a minimum of 2 min using a syringe pump (Harvard Apparatus), whereby flow rates were controlled by syringe size to maintain a 4:1:4 ratio of the stream widths. In an identical manner to the MCD, a chemostimulus gradient develops in the channel via diffusion, and the chemotactic response of the cell population was observed over time. Imaging was performed with phase-contrast microscopy (4×, 0.13 NA objective; Nikon Ti-E) at 1 fps over the course of approximately 10 min using a CMOS camera (Blackfly S, Teledyne FLIR). Sample size in each replicate chemotaxis experiment ranged from 7000 to 16,000 measured cell positions, dependent on the strain used for each particular assay.

## Experiment replicates and device reusability

After completing the initial 10 min filming period corresponding to the first measurement of a given organism/chemostimulus pairing, technical replicates were achieved in both the MCD and SA devices by restarting the flow. Stratification was maintained for at least 2 min to ensure consistent initial conditions among replicates. Next, the flow was halted and recording of bacteria positions commenced. This process was repeated for all subsequent technical replicates. A new MCD was fabricated for each unique organism/chemostimulus combination tested to prevent any cross-contamination between assays. Between biological replicates for a given organism/chemostimulus pairing, the MCD was cleaned by first flowing ethanol and then deionized water through all of the inlets. The device was then dried by flowing clean compressed air through the device and placing it under vacuum. Finally, the MCD was pre-treated again with a 0.5% (w/v) BSA solution and flushed with ASW prior to additional experiments. With appropriate cleaning, the same MCD can safely be used for multiple biological replicates of the same organism/chemostimulus combination. It is recommended to use a new device outside biological replicates of a particular organism/chemostimulus combination, consistent

with the majority of PDMS applications (*Toepke and Beebe, 2006*). For SA experiments, a new device was used for each biological replicate in *Figure 2e*, although SA devices can also be easily cleaned and reused in the same manner as the MCD.

## TEM imaging

For each species, initial cultures were grown following the previously described protocols (without any initial washing/resuspending), before the following final cell suspensions were prepared: (i) 4 ml of *V. alginolyticus* culture washed and resuspended (1,500 relative centrifugal force (RCF) for 5 min) in 1 ml of fresh 2216 media, (ii) 1 ml of *P. haloplanktis* culture washed and resuspended (1200 RCF for 5 min) in 1 ml of fresh 2216 media, then diluted 10× in DDW (double distilled water), and (iii) 8 ml of *E. coli* culture washed and resuspended (4000 RCF for 5 min) in 1 ml of DI water, then diluted 10× in DDW. For each species, 4 µl of cell suspension was applied to a glow discharged copper mesh carbon coated grid and allowed to adsorb to the grid for 30 s. The grids were briefly washed in DDW, followed by staining with 1% Aqueous Uranyl Acetate, and allowed to dry fully before imaging. Grids were imaged using a FEI Morgagni transmission electron microscope (FEI, Hillsboro, OR) operating at 80 kV and equipped with a CMOS camera (Nanosprint5, AMT).

## Acknowledgements

We thank J Vlahakis of the Tufts Micro and Nano Fabrication Facility for assistance in the device fabrication, JB. Noble for preliminary work on the microfluidic device design, and R Stocker and M Salek for helpful discussions. TEM samples were prepared and imaged by the Brandeis Electron Microscope Facility. This work was funded by NSF Awards OCE-1829827, CAREER-1554095, and CBET-1701392 (to JSG), and OCE-1829905 (to SAF).

## Additional information

### Funding

| Funder | Grant reference number | Author |
| --- | --- | --- |
| National Science Foundation | 1829827 | Michael R Stehnach Richard J Henshaw Jeffrey S Guasto |
| National Science Foundation | 1554095 | Michael R Stehnach Richard J Henshaw Jeffrey S Guasto |
| National Science Foundation | 1701392 | Michael R Stehnach Richard J Henshaw Jeffrey S Guasto |
| National Science Foundation | 1829905 | Sheri A Floge |

The funders had no role in study design, data collection and interpretation, or the decision to submit the work for publication.

### Author contributions

Michael R Stehnach, Richard J Henshaw, Conceptualization, Data curation, Software, Formal analysis, Validation, Investigation, Visualization, Methodology, Writing – original draft, Writing – review and editing; Sheri A Floge, Conceptualization, Supervision, Funding acquisition, Investigation, Writing – original draft, Writing – review and editing; Jeffrey S Guasto, Conceptualization, Supervision, Funding acquisition, Investigation, Methodology, Writing – original draft, Project administration, Writing – review and editing

### Author ORCIDs

Michael R Stehnach https://orcid.org/0000-0003-0302-4641
Richard J Henshaw https://orcid.org/0000-0001-7053-9341
Sheri A Floge https://orcid.org/0000-0002-1996-3347

Jeffrey S Guasto https://orcid.org/0000-0001-8737-8767

**Decision letter and Author response**
Decision letter https://doi.org/10.7554/eLife.85348.sa1
Author response https://doi.org/10.7554/eLife.85348.sa2

## Additional files

### Supplementary files
• MDAR checklist

### Data availability
Data files used during the current study, including all CAD designs and drawings, and code to reproduce the analysis reported are publicly available at BCO-DMO (https://www.bco-dmo.org/dataset/885701). The algorithm codes are described in the Materials and methods.

The following dataset was generated:

| Author(s) | Year | Dataset title | Dataset URL | Database and Identifier |
| --- | --- | --- | --- | --- |
| Henshaw RJ, Stehnach MR, Floge SA, Guasto JS | 2023 | Multiplexed microfluidic screening of bacterial chemotaxis | https://www.bco-dmo.org/dataset/885701 | Biological and Chemical Oceanography Data Management Office, 885701 |

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

## Appendix 1

### Multiplexed chemotaxis device (MCD) microchannel resistances and dimensions

Hydraulic circuit analysis (Materials and methods) was used to determine the hydraulic resistance and thus geometry of the individual microchannels comprising the MCD channel network (*Figure 1d and e*). The location of each channel in the dilution layer is indicated in the device layout in *Figure 1d* and in the hydraulic circuit in *Figure 1—figure supplement 2a*. Subscripts "CIL" indicate channels in the cell injection layer shown in *Figure 1e* and *Figure 1—figure supplement 2b*. The resistance for the observation region ($R_{4,4}$) varies in the microchannel width due to the hydraulic resistance varying between the channel sections before and after the chemical and buffer solution meet (*Figure 1d*), and the micromixer ($R_M$) has nonuniform channel height due to the herringbone ridges (Materials and methods; see also *Figure 1—figure supplement 1*).

**Appendix 1—table 1.** Multiplexed chemotaxis device (MCD) microchannel resistance and dimensions (corresponding to *Figure 1—figure supplement 2f*).

| Channel | Resistance (mPamPa.s. μm-3) | Height (μm) | Width (μm) | Length (mm) |
|---|---|---|---|---|
| $R_{0,2}$ | 0.0082 | 90 | 90 | 19.06 |
| $R_{1,1}$ | 0.0095 | 90 | 90 | 22.08 |
| $R_{2,1}$ | 0.0147 | 90 | 90 | 34.06 |
| $R_{3,1}$ | 0.0201 | 90 | 90 | 46.43 |
| $R_{4,1}$ | 0.0277 | 90 | 90 | 64.13 |
| $R_{5,1}$ | 0.0300 | 90 | 90 | 69.45 |
| $R_{0,4}$ | 0.0201 | 90 | 90 | 46.50 |
| $R_{1,4}$ | 0.0149 | 90 | 90 | 34.54 |
| $R_{2,4}$ | 0.0098 | 90 | 90 | 22.60 |
| $R_{3,4}$ | 0.0046 | 90 | 90 | 10.76 |
| $R_B$ | 0.0035 | 90 | 150 | 19.93 |
| $R_M$ | 0.0043 | - | 200 | 40.7 |
| $R_{4,4}$ | $7.7 \times 10^{-4}$ | 90 | - | - |
| $R_{CIL,1}$ | 0.0575 | 75 | 75 | 64.24 |
| $R_{CIL,2}$ | 0.0142 | 75 | 75 | 15.89 |

