## [Editor Report]

This manuscript presents a valuable new microfluidic tool that will allow researchers from different fields to rapidly quantify the chemotactic response of microbes to chemical gradients that have different strengths. Using planktonic bacteria, this paper convincingly shows that a multiplexed microfluidic device produces similar results to previously described microfluidic devices that generate only one gradient at a time. By performing on-chip dilutions, this device allows data for six different gradient strengths to be generated simultaneously, potentially reducing both experimental effort and biological variability.

---

## [Decision Letter]

**Decision letter after peer review:**

Thank you for submitting your article "Multiplexed microfluidic screening of bacterial chemotaxis" for consideration by *eLife*. Your article has been reviewed by 2 peer reviewers, and the evaluation has been overseen by a Reviewing Editor and Meredith Schuman as the Senior Editor. We regret the very long time it has taken to complete the review process – we had to consult a number of reviewers before obtaining the reviews below.

Essential revisions:

General matters:

1) We urge the authors to make available the original channel designs they used for these 2-layer devices, e.g. 3D CAD files or similar, to make it as easy as possible for another user to replicate/print this design.

2) Please also make available any datasets and codes associated with this project, including any custom codes for calibrating the channels.

3) In the discussions, expand on the vision of what exactly these devices might be useful for, beyond the type of study demonstrated in Figure 4, perhaps beyond bacterial chemotaxis (e.g. what about eukaryotic chemotaxis) and with some discussion of design limitations.

Specific issues:

1. In several of the examples presented, e.g. Figure 2a, g, it seems that at long times (>5 minutes) a significant negative chemotactic response follows from the initially positive chemotactic response (not just delayed accumulation), where does this come from? Is this a known effect?

2. Related to the above comment, it seems that a major advantage of the MCD is its ability to sustain precise chemical gradients over a spectrum of concentrations, so these results could provide new insights into the dynamics of bacterial chemotaxis and accumulation, as demonstrated in the curves of β(t) vs time in figures 2,3. Perhaps Figure 4 could be supplemented with some more sophisticated analysis of the dynamics of the response – how does β vary in time at the different concentrations, not just the behaviour of max-β.

3. A few more words of justification on the choice of species/chemical would be helpful – why select those combinations? The response of VA to leucine is not unexpected, given observations in other motile species. It would have been more interesting to see the MCD applied to a greater number of species/chemostimulant than these standard combinations (for comparison, the scope of Raina et al., 2022, was much more extensive). Alternatively, could there be some attempt to connect – at a mechanistic or molecular level – these detailed measurements of how the cells responded to chemoattractants to their species-specific motility strategies/number of flagella, etc?

4. What determines the width of the central band of immotile cells that is to be ignored in the analysis? Was this always the same width across different species? Is anything gained by removing the band from the analysis? What if this procedure introduces some bias by removing motile bacteria that are transitioning from a positively chemotactic towards a negatively chemotactic state?

5. The concluding paragraphs seem to be somewhat self-congratulatory, while there are many potential applications of the device, the paper only demonstrated one scenario where it concretely outperforms traditional assays in terms of efficiency, the usefulness and relevance of the MCD for 'ecosystem and human health investigations' are grand claims that remain to be tested.

6. For balance, there should also be some statements outlining some of the limitations of the device, how accessible is this design for a lab ill-equipped to build multilayer devices compared to the traditional single assay designs, what about the propensity for clogging, etc.

7. Line 56: change principle to principal.

8. Reconsider the use of "stratified", "stratification" etc. Using "fluid stratification" to describe chemical gradients does not seem like the most straightforward word choice, however, is this is a stylistic choice?

9. It is noted that these devices are "reusable"? Obviously, one might be apprehensive about reusing a device that has been exposed to phenol or other toxins, or other bacterial species that might affect motility of the focal strain. Were the devices actually reused in this study? If so, maybe you can explain how they were "cleaned" between uses? e.g. by injecting ethanol or..?? This might be useful for users of these devices.

10. Figure 2I – please describe how the constant gradients strengths were obtained using the device. It is not clear if these measurements were obtained using a multiplexed device operated in a different manner or..?? The description provided in Figure 1 —figure supplement 3 is not clear to us.

11. Figure 3. – all of the cells in the central 250 µm wide band are not necessarily "non-motile" (a fraction of these are certainly motile given the random nature of bacterial motility). The same in Line 208 of the manuscript. Also, it would be more straightforward to list the bacterial strain and chemoaffector on the figure itself to avoid having to dig so deeply into the caption (especially since you show an EM image of each strain).

12. Line 281: it is not clear what you mean by "alluded"?

13. Line 323: rather than "aging cells" it would be more accurate to say something like "metabolic changes in cell cultures that occur over time".

14. Line 328: how might the MCD design be amenable to cell retrieval (presumably you mean isolating chemotactic cells from non-chemotactic ones), given that the channels have a common outlet?

15. Line 570: it is noted that "Each experiment was technically repeated at least three times with the same culture and repeated on different days with freshly grown cells." However, it is not clear what the error bars and the shaded regions around line graphs in the various figures actually indicate. Are these indicating variability across technical repeats or biological repeats or..?

16. Figure 1 —figure supplement 3. It is noted that: "To achieve symmetric flow the applied pressure was tuned." This tuning was required because of the small variations in channel height and tubing lengths. Is such tuning required to ensure that the desired dilutions are accurate? It seems that such tuning could only be accomplished with dye in a separate "calibration" experiment. As noted in the public review, it is not clear whether these devices can produce accurate dilutions "off the shelf" or whether the applied pressures of each device need to be calibrated in separate experiments with dye.

17. Figure 2 —figure supplement 2: sentence after (b) is lacking a verb.

---

## [Author Response]

Essential revisions:General matters:1) We urge the authors to make available the original channel designs they used for these 2-layer devices, e.g. 3D CAD files or similar, to make it as easy as possible for another user to replicate/print this design.

The original channel designs have been submitted to an existing dataset in the BCO-DMO database and will be publicly available imminently. A citation (line #733) for this dataset has been included in the data availability statement for ease of access.

2) Please also make available any datasets and codes associated with this project, including any custom codes for calibrating the channels.

All codes and supporting material, including custom codes for channel calibrations, have been submitted to the BCO-DMO database and will be publicly available imminently. A citation (line #733) for this dataset has been included in the code availability statement for ease of access.

3) In the discussions, expand on the vision of what exactly these devices might be useful for, beyond the type of study demonstrated in Figure 4, perhaps beyond bacterial chemotaxis (e.g. what about eukaryotic chemotaxis) and with some discussion of design limitations.

We thank the Reviewers for this feedback – we agree that our original Discussion was limited on other potential research avenues to which this device might be applied, as well as limitations of the device itself. As mentioned in our original Discussion, the current design will accommodate eukaryotic cells and could be scaled up for larger organisms, but we have now expanded our revised Discussion (along with incorporating comments from the Public Reviews) to include how the MCD could be adapted to study other conditions such as slow steady flow for surface-attached microorganisms for example. Furthermore, a new paragraph has been added to the Discussion to comment on some of the limitations of the MCD (again, incorporating other comments from Specific Issues raised in these reviews) and how such limitations could be mitigated for potential future users (line #348).

Specific issues:1. In several of the examples presented, e.g. Figure 2a, g, it seems that at long times (>5 minutes) a significant negative chemotactic response follows from the initially positive chemotactic response (not just delayed accumulation), where does this come from? Is this a known effect?

Yes, for particularly high initial concentrations, the direction of chemotactic bias changes in the channel, which was briefly noted in the Results section (“Validation of MCD performance….”) of our original manuscript. Specifically, this is a consequence of the serine consumption of V. alginolyticus – as the cells migrate up the gradient they are also consuming the available serine. The high degree and rapid nature of their accumulation in the high concentration region results in a rapid consumption of serine. This local depletion of serine combined with the diffusion of remaining serine across the channel can result in an inversion of the chemical gradient and thus a reversal of the cell accumulation at intermediate times. This phenomenon is not observed at lower initial concentrations, because after any initial consumption, the serine concentrations (and thus gradients) are likely below the detection limits of V. alginolyticus. However, regardless of the mechanism, we observe highly consistent behaviour between the single assay (SA) device and the MCD (Figure 2e-j). From a purely microfluidic perspective, the performance of the two devices is highly consistent with one another under these particular conditions (i.e. apart from any nuances of the biological response of the cells). Based on the Reviewer’s comment, we now recognize that our description of this behavior in the original manuscript was too brief and insufficient for the reader. In response, we have now expanded on this description in the Results section (line #168; “Validation of MCD performance….”) to improve the clarity surrounding this point.

2. Related to the above comment, it seems that a major advantage of the MCD is its ability to sustain precise chemical gradients over a spectrum of concentrations, so these results could provide new insights into the dynamics of bacterial chemotaxis and accumulation, as demonstrated in the curves of β(t) vs time in figures 2,3. Perhaps Figure 4 could be supplemented with some more sophisticated analysis of the dynamics of the response – how does β vary in time at the different concentrations, not just the behaviour of max-β.

We thank the Reviewer for this feedback, and agree there is scope for more sophisticated analysis of the accumulation dynamics from assays such as the MCD. While we emphasize that the primary goal of the present work is to demonstrate the performance of the device, in response to this feedback we have carried out several additional analyses into the accumulation dynamics captured by β(*t*). These results are now included in an updated Figure 4 (top, page 9) and associated text (line #289).

To supplement our original analysis, we examined the rate of response by calculating the maximal value of *d*β/*dt* (prior to peak accumulation), which is indicative of the rate of accumulation and the chemotactic velocity of the bacteria (Son et al. 2016). The rate of response is a particularly useful metric to separate the strong observed responses of different species. In particular, at high concentrations (200 uM), V. alginolyticus and P. haloplanktis exhibit a similar degree of accumulation (β_max_) to serine along with V.

alginolyticus to leucine. In contrast, the maximal rate of accumulation max(*d*β/*dt*) for the rapid-swimming P. Haloplanktis is nearly twice as high as V. alginolyticus at 200 uM. Separately, for chemotaxis of V. alginolyticus to leucine, these bacteria exhibit a maximum degree of accumulation at 49 μm but accumulate fastest at lower chemostimulant concentrations of 3.6 uM.

To address this feedback, we have added a new Figure 4 (b) (top, page 9) showing the maximal response rate (i.e. max(*d*β/*dt*), prior to the maximum cell accumulation) along with associated text in the Results section (line #289; “Multiplexed microfluidic device …”). As pointed out by the Reviewer, this additional analysis provides a broader perspective on the responses summarised by β(*t*). This approach demonstrates an advantage of the MCD in comparison to other high-throughput assays in being able to extract cell motility information from such single-cell assays in addition to simple population accumulation. We envision that future applications will take advantage of the flexibility afforded by the MCD and will adapt and improve upon the analytical approaches presented here.

3. A few more words of justification on the choice of species/chemical would be helpful – why select those combinations? The response of VA to leucine is not unexpected, given observations in other motile species. It would have been more interesting to see the MCD applied to a greater number of species/chemostimulant than these standard combinations (for comparison, the scope of Raina et al., 2022, was much more extensive). Alternatively, could there be some attempt to connect – at a mechanistic or molecular level – these detailed measurements of how the cells responded to chemoattractants to their species-specific motility strategies/number of flagella, etc?

The choices of species/chemical were selected due to their prominence in chemotaxis literature – this is briefly summarised in the beginning of the Results section “Multiplexed microfluidic device supports high-throughput chemotaxis screening for novel stimuli and various microorganism”. The chemotaxis of the organisms in the present study to model attractants such as serine has been well-documented and is extremely robust. Between the three organisms, we cover two main branches of bacterial motility: the classical run-tumble of *E. coli*, and the run-reverse-flick of both P. haloplanktis and V. alginolyticus. Hence, these species/chemicals were chosen primarily for benchmarking the MCD against a single gradient generation assay (SA) due to the (i) robustness, (ii) reliability, and (iii) reproducibility of the known chemotactic response. This approach aided in eliminating (as much as possible) other sources of uncertainty between the two devices.

The Reviewer is indeed correct – the MCD does open up interesting applications including (but not limited to) a larger range of species/chemostimulant and motility-dependent studies. We appreciate the Reviewer pointing out Raina et al. 2022, which was considerably more extensive in its application. In their work, the authors applied the in-situ chemotaxis assay (ISCA) to a large multi-year field study to study the chemotactic responses of prokaryotes towards phytoplankton-derived dissolved organic matter (DOM) exudates. However, we emphasise that a fairer comparison would perhaps be the original publication (Lambert et al., Nat. Microbiol. (2017); cited in main text) of the ISCA device, without which the study in Raina et al. 2022 would not have been possible. In Lambert et al. 2017, they first establish a method, then demonstrate the effectiveness of the ISCA device in a laboratory setting. To accomplish this, they rely on the known chemotactic response of Vibrio coralliilyticus, before undertaking a first in-situ deployment. The work by Lambert et al. 2017, thus opened the door to a range of studies, which extensively utilise the ISCA device, including Raina et al. 2022.

In our current work, we are proposing the MCD as a complementary approach to such devices as the ISCA – i.e. to provide a high-throughput microfluidic alternative that can both rapidly screen across concentrations whilst retaining the ability to extract detailed information from both the population and single-cell levels. We do not doubt that future studies will be able to apply the MCD to a more extensive range of conditions and combinations now that the device has been established. However, our ability to make compelling, broad insights through comparative studies among the organisms tested – based on motility patterns or flagellation – is unfortunately beyond the scope of the present work. We feel that such comparisons would be largely speculative and divergent from our primary goal of establishing the performance of the MCD.

To address the above points based on this Reviewer comment, we have added text and supporting references to the end of the aforementioned Results section (line #297; “Multiplexed microfluidic device supports …”) and Discussion section highlighting the complementary nature of the MCD in comparison to existing techniques such as the ISCA (line #297).

4. What determines the width of the central band of immotile cells that is to be ignored in the analysis? Was this always the same width across different species? Is anything gained by removing the band from the analysis? What if this procedure introduces some bias by removing motile bacteria that are transitioning from a positively chemotactic towards a negatively chemotactic state?

The initial (t=0 min) band of cells in the centre of the device includes all cells – both motile and non-motile (Figure 2e,g). The width of this initial band is set by the predetermined flow rates in the three branches (Figures 1a and 2b,c) leading into the observation region of the device (Figure 1 —figure supplement 3). While the fraction of motile cells is different across species, the initial spatial distribution of cells across species was constant in the device – provided that the flow rates were sufficient to maintain stratification. We agree this point was not clear in the original manuscript, so we have now added an additional figure supplement (specifically Figure 3 Figure Supplement 1) which shows the cell distributions at the beginning (t=0 min; solid lines) and end (t=10 min; dashed lines) of each experiment for all observation channels in the MCD. At time t=0 (a), the cell positions from all experiments consistently overlap in the expected zone in the centre of the channel. At time t=10 min (b), the motile cells have dispersed with the exception of the sub-population of non-motile cells, which remain in the centre of the channel. This feature is more prominent, for example, with P. haloplanktis than it is with V. alginolyticus. The vertical dashed black lines indicate the 250 μm wide “excluded” region of cells corresponding to the central band. The spatial distribution of non-motile cells was consistent across all experiments and comparable to the theoretical width (111um) based on the relative flow rate of the cell injection stream. Based on this high degree of consistency, we chose a fixed exclusion zone of 250 μm in the channel centre to encapsulate the majority of non-motile cells for all experiments.

We emphasise that the cells in the central band are not entirely ignored and that our choice of exclusion region has no bearing on the outcome of our analysis. The benefit of excluding this region from the conditional probability plots (P(y|t); Figures 2e,g and 3a,c,e,g) is that it better highlights the chemotactic nature of the cells: Large numbers of non-motile cells simply skew the visualisation by drawing the reader’s attention to strong features in the plots, which represent cells that are not contributing to the population level response. The calculation of β(*t*) only depends upon cells that approach within 200 μm of the upper/lower boundaries (Figure 2c), which is a typical method of studying cell accumulation (Seymour et al. 2010, Stocker et al. 2008). Thus, the instantaneous cell locations in the central band and those used in the calculation of β(*t*) are mutually exclusive, with the former having no bearing on β(*t*). We further note that any cells initially in the central band, which leave that region are of course counted in the analysis if they migrate toward the upper/lower boundaries.

We now clarify all of these points in relevant sections of the revised manuscript (Figure 3 caption (page 7), Results: Validation of MCD performance against conventional chemotaxis assays). See also our response to Specific Issue #11 below.

5. The concluding paragraphs seem to be somewhat self-congratulatory, while there are many potential applications of the device, the paper only demonstrated one scenario where it concretely outperforms traditional assays in terms of efficiency, the usefulness and relevance of the MCD for 'ecosystem and human health investigations' are grand claims that remain to be tested.

We agree with the Reviewer that we do lean heavily into applications outside of the tested scope of the MCD. We have toned down our language in the concluding paragraphs and added literature support where appropriate to better place the MCD in context of potential future applications. Additionally, we have added some discussion of limitations of the MCD in this section per Specific Point #6 (see response below).

6. For balance, there should also be some statements outlining some of the limitations of the device, how accessible is this design for a lab ill-equipped to build multilayer devices compared to the traditional single assay designs, what about the propensity for clogging, etc.

We thank the Reviewer for this feedback, and we agree that our original Discussion did not sufficiently balance the limitations of the device against its advantages. To address this, we have added a new paragraph to the Discussion section (line #348; also mentioned in our reply to General Point #3) to summarise some limitations of the device, and how these might be mitigated by potential future users of the MCD.

To the Reviewers’ specific points: We expect that most laboratories capable of conventional single layer microfluidics will also have the capability to produce multilayer devices with multilayer lithography, where the key piece of equipment is a mask aligner. In our experience, neither conventional single assay devices nor the MCD are prone to clogging by cells during normal operation. However, partial or full blockages of channels may occur during the fabrication process or if contaminants are present in the cell media or buffer (i.e. dust). Thus, normal precautions should be taken as for any microfluidics experiments, including inspecting devices after fabrication and filtering media to the extent possible. We now comment directly on both of these points and additional potential issues in both the Discussion (line #363) and Methods (“line #496”) sections of the revised manuscript.

7. Line 56: change principle to principal.

This typo has been corrected, and the manuscript has been proofed for further grammatical errors.

8. Reconsider the use of "stratified", "stratification" etc. Using "fluid stratification" to describe chemical gradients does not seem like the most straightforward word choice, however, is this is a stylistic choice?

We thank the reviewers for pointing out the potential confusion with the phrasing. We emphasize that the word “stratification” is used to describe only the initial state of the gradient, which is set by the very strong initial fluid flow. In other words, the initial fluid flow actively maintains three distinct regions having abrupt interfaces within the observation channel. We feel that this intended use is consistent with the general definition of the word stratum, meaning a layer or series of layers (i.e. not meant to be taken as “density stratified” in the traditional fluid mechanical sense). Where the word “stratified” is first mentioned in both the main text and Figure 1, we clarified the wording in hopes to indicate the word stratified explains the pattern formed by the 3 fluids flowing. For example, the first paragraph of the Results section now reads (line #106):

“The individual assays in the observation region are based on laminar flow patterning and established stop-flow diffusion methods: Rapid, parallel flow of chemostimulus (concentration, $C_i$) and buffer ($C=0$) solutions maintain initially stratified fluid regions. Upon stopping the flow, a chemostimulus gradient forms via diffusion (Figure 1a,b).”

9. It is noted that these devices are "reusable"? Obviously, one might be apprehensive about reusing a device that has been exposed to phenol or other toxins, or other bacterial species that might affect motility of the focal strain. Were the devices actually reused in this study? If so, maybe you can explain how they were "cleaned" between uses? e.g. by injecting ethanol or..?? This might be useful for users of these devices.

The Reviewers are correct that users should proceed with caution when using any device that previously contained incompatible materials. However, the MCD devices and most other PDMS microfluidic devices (e.g. the single gradient generator devices also employed here) can be reused in a number of ways: First, as demonstrated in the present study, technical replicates can be easily achieved by simply restarting the flow, allowing the fluid streams to re-stratify, then stopping the flow to begin a new replicate experiment. Second, if the chemoaffectors/organisms are identical, devices can be reused between different biological samples by cleaning the device, which involves flowing ethanol, then DI water sequentially through all of the inlets. Devices are then dried by flowing clean compressed air through the devices and finally placed under vacuum. Ethanol readily sterilizes the surfaces and helps to desorb cells and chemoaffectors from the surfaces. It is highly miscible with water and highly volatile, allowing it to be efficiently washed and evaporated from the device, respectively.

We agree this reusability is not clear in the manuscript and is useful information for future users of the MCD (as well as being an advantage of such microfluidic designs). Hence, we have added a subsection to the Methods (line #618; “Experiment replicates and device reusability”) summarising how replicates were achieved and the cleaning protocol for the device. Additionally, we added the following statements for future users to consider:

“…caution should be exercised to ensure no bias is introduced. It is recommended to use a new device outside biological replicates of a particular organism/chemostimulus combination, consistent with the majority of PDMS applications.”

10. Figure 2I – please describe how the constant gradients strengths were obtained using the device. It is not clear if these measurements were obtained using a multiplexed device operated in a different manner or..?? The description provided in Figure 1 —figure supplement 3 is not clear to us.

We apologise for any confusion around Figure 2l (page 6). We have revised this figure caption for clarity, which now reads:

“No gradient (k) conditions ($C_i=0$) were obtained by injecting buffer into the chemical inlet (setting $C_0=0$). Fixed gradient (l) conditions (Ci=200 μM of serine) were obtained by injecting C0=200 μM of serine into both the chemical and buffer inlets of the dilution layer.”

A similar description is also provided in the caption for Figure 2 —figure supplement 1, from which Figure 2l was derived.

Additionally, we have revised the caption for Figure 1—figure supplement 3 (line #992) for clarity.

11. Figure 3. – all of the cells in the central 250 µm wide band are not necessarily "non-motile" (a fraction of these are certainly motile given the random nature of bacterial motility). The same in Line 208 of the manuscript. Also, it would be more straightforward to list the bacterial strain and chemoaffector on the figure itself to avoid having to dig so deeply into the caption (especially since you show an EM image of each strain).

The Reviewer is correct in the split between motile/non-motile sub-populations in the central region of the channel: Initially (t=0 min), all cells including both motile and non-motile cells are located in the central band. As the motile bacteria migrate relative to the chemoaffector and swim randomly, they mostly disperse from the central region. In this work we refer to non-motile cells as those which have not moved (aside from diffusion) over the course of an experiment. See also our response to Specific Point #4 and the new figure addition of Figure 3 —figure supplement 1, which shows a consistent sub-population of cells that do not move from their starting positions for the full duration of each experiment. Undoubtedly, this band still contains some motile cells. However, we emphasise that this band is omitted simply for clarity in the kymographs of P(y|t). As such, this has no impact on our subsequent analysis, as β(t) by definition is computed using cell subpopulations far outside of that central band.

To further improve the clarity around this 250 μm wide central band, we have revised the Figure 3 (page 7) caption to read:

“Central 250 μm wide band, which often contains a significant subpopulation of non-motile cells, is omitted from P(y|t) for clarity with no impact on β(t).”

We have also added text to the manuscript (“Validation of MCD performance….”) that explicitly reads (line #210):

“The persistent central band of cells at later times is due to a sub-population of bacteria which remain non-motile over the course of each assay and are thus localised near y=0 (Figure 2e,g; Figure 3—figure supplement 1). Because any nonmotile or motile bacteria in this region do not impact the calculation of β(t), the central band is omitted from future kymographs for visualization purposes.**”**

Finally, we have updated Figure 3 to include the strain names and chemoaffectors as titles on the accumulation plots (Figures 3bdfh), as well as the new Figure 3 —figure supplement 1 (line #885).

12. Line 281: it is not clear what you mean by "alluded"?

The word “alluded” has been omitted, and the sentence has been revised to read “… a feature that is consistent with the chemotactic sensitivity of *E. coli* to serine …”.

13. Line 323: rather than "aging cells" it would be more accurate to say something like "metabolic changes in cell cultures that occur over time".

The phrase “aging cells” has been revised to “changes in cell cultures that may occur over time”.

14. Line 328: how might the MCD design be amenable to cell retrieval (presumably you mean isolating chemotactic cells from non-chemotactic ones), given that the channels have a common outlet?

Our choice of a single common outlet for this particular iteration of the MCD design stemmed from our focus on demonstrating the primary mode of operation of the device. However, the device design could be easily altered to accommodate separate outlets for each of the individual channels. Moreover, sophisticated existing chemotactic cell-sorting designs may be implemented for each channel to separate and collect subpopulations of cells with different chemotactic performance (e.g. Mao et al. 2003). We agree that this aspect was not clearly articulated in the manuscript, and thus we have updated portions of the text in the Discussion to highlight this (line #331), including supporting citations.

15. Line 570: it is noted that "Each experiment was technically repeated at least three times with the same culture and repeated on different days with freshly grown cells." However, it is not clear what the error bars and the shaded regions around line graphs in the various figures actually indicate. Are these indicating variability across technical repeats or biological repeats or..???

We apologise for this lack of clarity, and we have now revised the description of the error bars/shading in the appropriate figure captions. Variability is reported in most instances across biological repeats.

16. Figure 1 —figure supplement 3. It is noted that: "To achieve symmetric flow the applied pressure was tuned." This tuning was required because of the small variations in channel height and tubing lengths. Is such tuning required to ensure that the desired dilutions are accurate? It seems that such tuning could only be accomplished with dye in a separate "calibration" experiment. As noted in the public review, it is not clear whether these devices can produce accurate dilutions "off the shelf" or whether the applied pressures of each device need to be calibrated in separate experiments with dye.

We thank the reviewer for the comment. Foremost, we emphasise that the accuracy of the dilution is set by the design and fabrication of the dilution layer alone (Figure 1c,d), and it is independent of the calibration mentioned here. The thickness of the dilution layer channels is determined essentially by a single lithography process (Methods). The relative resistances of the dilution layer channels (Appendix 1; Figure 1—figure supplement 2) – and thus their dilution ratios – are largely unaffected by minor uncertainty in the absolute channel height from lithography.

However, differences in the fabricated height of the dilution layer versus the cell injection layer can lead to asymmetric flow of cells in the observation region (again, independent of the dilution process). The calibration to which the Reviewer refers is a simple process to ensure that the initial distribution of cells is centred in the observation channel during an assay (thus avoiding any bias in the cell distribution). Indeed, the device only requires a single calibration to determine the relative driving pressures required by the dilution and cell injection layers to achieve centering of the cells. Furthermore, this calibration need not be achieved with fluorescence microscopy or any specialised imaging: When setting up the experiment, observation of the cells (or tracer particles or dye) in the central channel are sufficient to perform pressure adjustment for calibration to ensure the cells start in the middle of the microchannel. In the present work, we used fluorescence imaging for demonstration purposes and ensure that the calibration was clear (Figure 1 —figure supplement 3). To clarify this issue, we have added the following text to emphasize the nature of the calibration and that it only needs to be performed once for the first device fabricated from a given set of molds:

– Results (“Multiplexed microfluidic device….”; line # 158):

“Due to variations in the replica mold fabrication process, a one-time tuning of the applied pressure ratio (p_{in,1-2} / p_{in,3-4}) for the dilution and cell injection layers is required to ensure symmetric flow in the observation channel (Methods).”

– Discussion (line #320):

“The MCD performs reliably for a wide range of applied pressures, and only requires calibration once, as long as the PDMS microchannels are cast from the same molds (Methods).”

– Methods (“MCD dilution, flow, and …”; line #532):

“The symmetry of the stratified chemostimulus and buffer distributions in the observation channel is critical to prevent bias in the chemotaxis measurements. As minor errors in the manufacturing process can alter this symmetry, the applied pressure to the cell injection layer ($p_{in},3,4$; Figure1—figure supplement 2) was tuned until the widths of the chemical, cell, and buffer streams in each observation channel were 4:1:4 ratio, respectively. Tuning was visualized by flowing a fluorescein solution (C_0 = 100 uM) in both the chemical and buffer inlets of the dilution layer as well as the buffer inlet of the cell injection layer (Figure 1d,e). The ratio of applied pressures (*p_in_;*1-2_*p_in_;*3-4) between the dilution and cell injection layer remained the same for all chemotaxis assays (*p_in_;*1-2_*p_in_;*3-4 = 10_7) which was slightly lower than the designed value (*p_in_;*1-2_*p_in_;*3-4 = 2). Tuning is only required for the first device fabricated from a particular set of molds, after which the calibration and tuning applies to all subsequent devices fabricated from the same mold set due to the robust nature of soft lithography. If significant fluid width variation across the observation regions is observed, this means the MCD is not functioning properly.”

17. Figure 2 —figure supplement 2: sentence after (b) is lacking a verb.

There is not a Figure 2 —figure supplement 2 in the original manuscript, so we believe this comment refers to either Figure 1 —figure supplement 2 or Figure 2 —figure supplement 1. We have reviewed the text in both captions and made grammatical corrections as noted by the Reviewers.